# Data-Dependent Regret Bounds
# for Constrained MABs

**Gianmarco Genalti**[*]
Politecnico di Milano
gianmarco.genalti@polimi.it

**Francesco Emanuele Stradi**[*]
Politecnico di Milano
francescoemanuele.stradi@polimi.it

**Matteo Castiglioni**
Politecnico di Milano
matteo.castiglioni@polimi.it

**Alberto Marchesi**
Politecnico di Milano
alberto.marchesi@polimi.it

**Nicola Gatti**
Politecnico di Milano
nicola.gatti@polimi.it

## Abstract

This paper initiates the study of *data-dependent* regret bounds in *constrained* MAB settings. These are bounds that depend on the sequence of losses that characterize the problem instance. Thus, in principle they can be much smaller than classical $\widetilde{\mathcal{O}}(\sqrt{T})$ regret bounds, while being equivalent to them in the worst case. Despite this, data-dependent regret bounds have been completely overlooked in constrained MABs. The goal of this paper is to answer the question: *Can data-dependent regret bounds be derived in the presence of constraints?* We provide an affirmative answer in constrained MABs with adversarial losses and stochastic constraints. Specifically, our main focus is on the most challenging and natural settings with *hard constraints*, where the learner must ensure that the constraints are always satisfied with high probability. We design an algorithm with a regret bound consisting of *two* data-dependent terms. The first one captures the difficulty of satisfying the constraints, while the second one encodes the complexity of learning independently of their presence. We also prove a lower bound showing that these two terms are *not* artifacts of our specific approach and analysis, but rather the fundamental components that inherently characterize the problem complexity. Finally, in designing our algorithm, we also derive some novel results in the related (and easier) *soft constraints* settings, which may be of independent interest.

## 1 Introduction

Over the past few years, constrained *multi-armed bandit* (MAB) problems have gained increasing popularity in learning theory (see, *e.g.*, [Liakopoulos et al., 2019, Pacchiano et al., 2021, Castiglioni et al., 2022a, Chen et al., 2022]). In unconstrained MAB problems, the learner is evaluated solely in terms of *regret*, which measures the difference between the learner's performance and the performance of an *optimal-in-hindsight* decision. Constrained settings introduce additional challenges, as the learner must also ensure that certain constraints are *not* violated excessively while learning.

A growing trend in unconstrained MAB research is the derivation of *data-dependent* regret bounds (see, *e.g.*, [Neu, 2015a, Lee et al., 2020a]). These bounds depend on the sequence of losses that

---

[*]Equal contribution.

39th Conference on Neural Information Processing Systems (NeurIPS 2025).

characterize the problem instance. Some example of such bounds—typically called *small-loss* regret bounds—are of the form $\widetilde{\mathcal{O}}(\sqrt{L^*})$, where $L^*$ denotes the cumulative loss of an optimal-in-hindsight decision. Clearly, data-dependent bounds can be much smaller than classical $\widetilde{\mathcal{O}}(\sqrt{T})$ regret bounds, while being equivalent to them in the worst case. Despite this, data-dependent regret bounds have been completely overlooked in the literature on constrained online learning problems.

The main goal of this paper is to initiate the study of data-dependent regret bounds in constrained MAB settings. In particular, we aim at answering the following research question:

*Can data-dependent regret bounds be derived in the presence of constraints?*

In this paper, we provide an affirmative answer to the question above, as described in the following.

We refer the reader to Appendix B for a complete discussion on related works.

## 1.1 Original Contributions

Given an impossibility result by Mannor et al. [2009], which prevents from obtaining sub-linear regret in constrained settings where *both* the losses and the constraints are selected *adversarially*, in this paper we focus on constrained MABs with *adversarial* losses and *stochastic* constraints, as customarily done in the literature (see, *e.g.*, [Qiu et al., 2020]). Specifically, our main focus is on the most challenging and natural settings with *hard constraints*, where the learner's goal is to minimize the regret while ensuring that the constraints are satisfied at every round with high probability.

### 1.1.1 Warm-Up: Soft Constraints

As a preliminary step toward our final goal, we design an algorithm with a *small-loss* regret bound in constrained MAB settings with *soft constraints*, which may be of independent interest. These settings only require that the *constraint violations* grow sub-linearly with the number of rounds $T$, thereby allowing to violate the constraints in some rounds. Although soft constraints are technically easier than hard ones, our algorithm incorporates some key components that are also central to the algorithm for hard constraints. At a high level, our algorithm builds upon an approach introduced by Lee et al. [2020a] to derive small-loss regret bounds in unconstrained problems. Our algorithm—called *Constrained OMD with Log-Barrier* (COLB)—applies this approach to a "safe" set of decisions that optimistically satisfy the constraints. This allows to attain sub-linear violations with a high-probability regret bound of $\widetilde{\mathcal{O}}(\sqrt{\sum_{t=1}^{T} \boldsymbol{\ell}_t^\top \boldsymbol{x}^*})$, where $\boldsymbol{x}^*$ is a decision that is optimal in hindsight while satisfying the constraints in expectation.

### 1.1.2 Hard Constraints

Our main result in this paper is an algorithm with a data-dependent regret bound in constrained MAB settings with *hard constraints*. Our algorithm for these settings—called *Safe OMD with Log-Barrier* (SOLB)—employs all the core components of COLB, and it adds new ones to deal with hard constraints. Specifically, SOLB always chooses a suitable combination between the decision suggested by COLB and a decision that *strictly* satisfies the constraints, which is given as input to the algorithm. The SOLB algorithm achieves a high-probability regret bound that is characterized by two data-dependent terms, described in the following.

- The first term is $\widetilde{\mathcal{O}}(\sqrt{\sum_{t=1}^{T} (\boldsymbol{\ell}_t^\top (\boldsymbol{x}^\diamond - \boldsymbol{x}^*))^2})$, where $\boldsymbol{x}^\diamond$ is the decision that strictly satisfies the constraints given as input. This term captures the difficulty due to constraints.

- The second term is $\widetilde{\mathcal{O}}(\sqrt{\sum_{t=1}^{T} \boldsymbol{\ell}_t^\top \boldsymbol{x}^*})$, and it intuitively encodes the performance of a decision $\boldsymbol{x}^*$ that is optimal in hindsight while satisfying the constraints in expectation.

Finally, we also provide a lower bound that demonstrates that the two terms in the regret bound of the SOLB algorithm are *not* artifacts of our specific approach and analysis, but rather the fundamental components that inherently characterize the complexities of the problem. Interestingly, this result also shows that data-dependent regret bounds can not only outperform classical $\widetilde{\mathcal{O}}(\sqrt{T})$ bounds, but also offer insights into the underlying complexities of learning problems.

## 2 Constrained Multi-Armed Bandits

In the *multi-armed bandit* (MAB) framework [Lattimore and Szepesvári, 2020], a learner is repeatedly faced with a decision among $K \in \mathbb{N}_+$ actions over $T \in \mathbb{N}_+$ rounds. At each round $t \in [T]$,[2] the learner chooses a strategy (*i.e.*, a probability distribution over actions) $\boldsymbol{x}_t \in \Delta_K$, where $\Delta_K$ is the simplex of dimension $K - 1$. Then, they play an action $a_t \sim \boldsymbol{x}_t$ sampled according to this strategy and observe a loss $\ell_t(a_t)$, which is defined by a vector $\boldsymbol{\ell}_t \in [0, 1]^K$ of losses at round $t$.[3] In this paper, we study a *constrained* version of the MAB framework [Pacchiano et al., 2021]. At each $t \in [T]$, in addition to a loss, the learner also observes $m \in \mathbb{N}_+$ constraint costs $g_{t,i}(a_t)$, one for each constraint $i \in [m]$. Each of them is determined by a vector $\boldsymbol{g}_{t,i} \in [0, 1]^K$ of constraint costs at round $t$. Each constraint $i \in [m]$ is associated to a threshold $\alpha_i \in [0, 1]$, and it is considered satisfied by a learner's strategy whenever the constraint cost is below $\alpha_i$ in expectation.

Motivated by a well-known impossibility result by Mannor et al. [2009],[4] in this paper we focus on constrained MAB problems in which the losses are chosen *adversarially* and the constraint costs are selected *stochastically*. Specifically, we assume that, at each round $t \in [T]$, the loss vector $\boldsymbol{\ell}_t$ is chosen by an *adaptive* adversary that is aware of the history of interaction up to round $t - 1$, while each cost vector $\boldsymbol{g}_{t,i}$, for $i \in [m]$, is sampled independently from a probability distribution $\mathcal{G}_i$. For ease of notation, in the following we use $\boldsymbol{\ell}_{1:T}$ to denote the sequence of all loss vectors $\boldsymbol{\ell}_t$, for $t \in [T]$, while we denote by $\boldsymbol{g}_i := \mathbb{E}_{\boldsymbol{g} \sim \mathcal{G}_i}[\boldsymbol{g}]$ the expected value of $\mathcal{G}_i$, for every $i \in [m]$.

The performance (in terms of losses) of a learning algorithm is usually measured in terms of regret with respect to a *baseline*. In the constrained MABs addressed in this paper, the baseline is formally defined by the following optimization problem parametrized by $\boldsymbol{\ell}_{1:T}$ and $\boldsymbol{g}_i$ for $i \in [m]$:

$$\text{OPT}(\boldsymbol{\ell}_{1:T}, \{\boldsymbol{g}_i\}_{i \in [m]}) := \begin{cases} \min_{\boldsymbol{x} \in \Delta_K} & \sum_{t=1}^{T} \boldsymbol{\ell}_t^\top \boldsymbol{x} \quad \text{s.t.} \\ & \boldsymbol{g}_i^\top \boldsymbol{x} \leq \alpha_i \quad \forall i \in [m]. \end{cases} \tag{1}$$

Program (1) encodes the value of a strategy that is *optimal in hindsight*, *i.e.*, a strategy that minimizes the cumulative loss while ensuring that the constraints are satisfied in expectation. In the following, we denote one such strategy, which is an optimal solution to Program (1), as $\boldsymbol{x}^* \in \Delta_K$. Then, for a sequence of losses $\boldsymbol{\ell}_{1:T}$, the *(cumulative) regret* over the $T$ rounds is defined as

$$R_T(\boldsymbol{\ell}_{1:T}) := \sum_{t=1}^{T} \boldsymbol{\ell}_t^\top \boldsymbol{x}_t - \text{OPT}(\boldsymbol{\ell}, \{\boldsymbol{g}_i\}_{i \in [m]}) = \sum_{t=1}^{T} \boldsymbol{\ell}_t^\top \boldsymbol{x}_t - \sum_{t=1}^{T} \boldsymbol{\ell}_t^\top \boldsymbol{x}^*.$$

**Remark 1** (On the regret definition). *Differently from the standard MAB framework [Lattimore and Szepesvári, 2020], in constrained settings an optimal strategy $\boldsymbol{x}^*$ may not coincide with a vertex of the simplex $\Delta_K$, i.e., an optimal action may not exist. This is intuitive since, whenever the action associated with the smallest loss in hindsight does not satisfy the constraints in expectation, an optimal strategy $\boldsymbol{x}^*$ may play that action as much as possible, while satisfying the constraints in expectation. This is the reason why the regret $R_T(\boldsymbol{\ell}_{1:T})$ is defined with respect to (randomized) strategies, rather than actions. This definition is standard in constrained online learning settings (see, e.g., [Efroni et al., 2020, Pacchiano et al., 2021, Bernasconi et al., 2022]).*

In this paper, our goal is to design learning algorithms for constrained MAB settings that achieve *small-loss style* regret bounds of the form $R_T(\boldsymbol{\ell}_{1:T}) \leq \widetilde{\mathcal{O}}(\sqrt{\sum_{t=1}^{T} \boldsymbol{\ell}_t^\top \boldsymbol{x}^*})$, where $\sum_{t=1}^{T} \boldsymbol{\ell}_t^\top \boldsymbol{x}^*$ represents the cumulative loss incurred by a strategy $\boldsymbol{x}^*$ that is optimal in hindsight. These regret bounds are arbitrarily better than common $\widetilde{\mathcal{O}}(\sqrt{T})$ regret bounds when $\boldsymbol{x}^*$ outperforms other strategies on the sequence of losses $\boldsymbol{\ell}_{1:T}$, while being equivalent to them in the worst case. It is well known that small loss regret bounds can be achieved in unconstrained MABs (see, *e.g.*, [Lee et al., 2020a]), but it remains an open question whether they can also be derived in constrained settings.

In constrained MABs, a fundamental challenge is the fact that the learner must account for constraint violations during learning. Our primary focus is on satisfying the constraints at every round with high probability. However, we also derive some results for the weaker goal of minimizing cumulative violations, as a preliminary step. Next, we formally introduce these two goals.

---

[2]Given $n \in \mathbb{N}$, we denote by $[n]$ the set $\{1, \dots, n\}$ of the first $n$ natural numbers.

[3]In this paper, we use the notation $c(i)$ to denote the $i$-th element of vector $\boldsymbol{c}$.

[4]Mannor et al. [2009] show that if *both* losses and costs are selected adversarially, *no* algorithm can simultaneously achieve sublinear (in $T$) regret and sublinear (in $T$) cumulative constraint violation.

**Soft Constraints** In this setting, the goal of the learner is to minimize the *(cumulative) positive constraint violations* over the $T$ rounds, defined as $V_T := \max_{i \in [m]} \sum_{t=1}^{T} \left[ \boldsymbol{g}_i^\top \boldsymbol{x}_t - \alpha_i \right]^+$, where we let $[\cdot]^+ := \max\{0, \cdot\}$. Intuitively, $V_T$ represents the total constraint violation accumulated by the learner during the learning process, and it also ensures that negative violations (*i.e.*, constraint satisfactions) do *not* cancel out positive ones. The goal is to guarantee that $V_T = o(T)$.[5]

**Hard Constraints** In this setting, the goal of the learner is to guarantee that $\boldsymbol{g}_i^\top \boldsymbol{x}_t \leq \alpha_i$ for every constraint $i \in [m]$ and round $t \in [T]$ with high probability.[6] This objective is only attainable under the following two assumptions, which are common in the literature on hard constraints settings (see, *e.g.*, [Pacchiano et al., 2021, Liu et al., 2021, Bernasconi et al., 2022]).

**Assumption 1** (Slater's condition). *Program (1) satisfies Slater's condition, i.e., it admits a strictly feasible solution $\boldsymbol{x}^\diamond \in \Delta_K$, which is a strategy such that $\boldsymbol{g}_i^\top \boldsymbol{x}^\diamond < \alpha_i$ for every constraint $i \in [m]$.*

**Assumption 2** (Knowledge of a strictly feasible strategy). *The learner knows a strictly feasible strategy $\boldsymbol{x}^\diamond := \arg\max_{\boldsymbol{x} \in \Delta_K} \min_{i \in [m]} [\alpha_i - \boldsymbol{g}_i^\top \boldsymbol{x}]$ and its associated cost $\theta_i := \boldsymbol{g}_i^\top \boldsymbol{x}^\diamond$.[7] We denote by $\rho \in [0, 1]$ to margin by which $\boldsymbol{x}^\diamond$ satisfies the constraints, formally $\rho := \min_{i \in [m]} [\alpha_i - \theta_i]$.*

Intuitively, Assumptions 1 and 2 are necessary to ensure that, in early rounds when little information is available, the learner has sufficient exploration opportunities without violating the constraints.

# 3 Being Safe While Learning With an Increasing Learning Rate

We begin by presenting two core components of our algorithms presented in Sections 4 and 5. The first one is a *safe decision space*, which is a restricted set of strategies used to control constraint violations. The second component is *OMD with log-barrier* [Lee et al., 2020a], which is an existing algorithm that achieves data-dependent regret bounds in unconstrained settings and serves as a foundation for our algorithms. The goal of this section is to show how these two components can be combined to achieve data-dependent regret bounds while controlling constraint violations.

## 3.1 Costs Estimation and Safe Decision Spaces

Estimating costs is a crucial task for any algorithm operating in constrained MABs. However, using these estimates in order to control constraint violations may *not* be trivial. Next, we describe the approach used by our algorithms. In the following, we let $N_t(a) := \sum_{\tau=1}^{t} \mathbb{1}_{\{a_\tau = a\}}$ be the number of rounds up to $t \in [T]$ in which action $a \in [K]$ is played. Then, an unbiased estimator for the cost of constraint $i \in [m]$ when playing action $a \in [K]$ is $\widehat{g}_{t,i}(a) := \frac{1}{\max\{1, N_t(a)\}} \sum_{\tau \in [t]} g_{\tau,i}(a) \mathbb{1}_{\{a_\tau = a\}}$. The following result quantifies the uncertainty associated with the estimator above.

**Lemma 1.** *Let $\delta \in (0, 1)$ and $\beta_t(a, \delta) := \min \left\{ 1, \sqrt{4 \ln(TKm/\delta) / \max\{1, N_t(a)\}} \right\}$ for $a \in [K]$. Then, with probability at least $1 - \delta$, $|\widehat{g}_{t,i}(a) - g_i(a)| \leq \beta_t(a, \delta)$ for every $t \in [T], i \in [m], a \in [K]$.*

This above lemma is a trivial consequence of Hoeffding's inequality and a union bound. We denote by $\widehat{\boldsymbol{g}}_{t,i} \in [0, 1]^K$ the vector whose entries are the estimates $g_{t,i}(a)$, while we let $\boldsymbol{\beta}_t(\delta) \in [0, 1]^K$ be the vector of the bounds $\beta_t(a, \delta)$. For clarity, in the rest of the paper we omit $\delta$ from the argument of $\boldsymbol{\beta}_t(\delta)$. Moreover, we denote by $\mathcal{E}(\delta)$ the event defined by Lemma 1, which satisfies $\mathbb{P}(\mathcal{E}(\delta)) \geq 1 - \delta$.

---

[5]Let us remark that, in the soft constraints setting, we are interested in obtaining small-loss bounds *only for the regret*. Indeed, small-loss bounds naturally belong to adversarial settings, as they become vacuous in stochastic ones (recall that constraint costs are stochastic in our setting). Moreover, an optimal-in-hindsight strategy $\boldsymbol{x}^*$ satisfies the constraints in expectation, by definition. Thus, it would *not* make any sense to derive a small-loss bound for the cumulative positive constraint violations $V_T$.

[6]A constraint $i \in [m]$ is satisfied at $t \in [T]$ whenever its cost is below $\alpha_i$ in expectation over the randomness of the strategy $\boldsymbol{x}_t$ and the cost $g_{t,i}$. This is standard in constrained MABs (*e.g.* [Pacchiano et al., 2021]).

[7]Previous works (see, *e.g.*, [Pacchiano et al., 2021, Liu et al., 2021, Bernasconi et al., 2022]) usually assume to know a generic strictly feasible strategy. For ease of presentation, in this work we assume to know a strategy that satisfies the constraints as much as possible. Nonetheless, our results can be generalized to the case commonly considered in the literature.

**Safe Decision Space**  Lemma 1 provides some confidence intervals for the estimated costs. Next, we describe how these intervals can be used to define a sequence of sets, called *safe decision spaces*, which contain strategies that an algorithm can employ to control the cumulative constraint violations, that is, attaining $V_T \leq \widetilde{O}(\sqrt{T})$. Formally, for every round $t \in [T]$, we let

$$\mathcal{S}_t := \left\{ \boldsymbol{x} \in \Delta_K : (\widehat{\boldsymbol{g}}_{t,i} - \boldsymbol{\beta}_t)^\top \boldsymbol{x} \leq \alpha_i \ \forall i \in [m] \right\}$$

be the *safe decision space* at time $t$. This definition is standard in the constrained MAB literature. Intuitively, an algorithm that selects strategies from $\mathcal{S}_t$ ensures that, with probability at least $1 - \delta$, the expected (optimistic) incurred costs remain below the thresholds at every round $t \in [T]$, and for every action $a \in [K]$ and constraint $i \in [m]$. It is easy to see that this holds thanks to Lemma 1 and the way in which the safe decision spaces are constructed.

### 3.2  Data-Dependent Bounds via Increasing Learning Rate

Next, we recall some needed details of the *OMD with log-barrier* algorithm by Lee et al. [2020a]. This algorithm achieves small-loss guarantees in unconstrained MABs, with a regret of $\widetilde{\mathcal{O}}(\sqrt{L^*})$ with high probability, where $L^*$ is the cumulative loss of an action that is optimal in hindsight. The algorithm works as standard OMD with log-barrier regularization $\psi_t(\boldsymbol{x}) = \sum_{a=1}^K \frac{1}{\eta_{t,a}} \ln \frac{1}{x(a)}$ (where $D_{\psi_t}(\cdot, \cdot)$ is the Bregman divergence built given $\psi_t$), where $\eta_{t,a}$ is an *increasing* sequence of learning rates. Specifically, each time the probability of selecting an action goes below a certain threshold, the learning rate is increased by a constant factor, while the threshold is increased. This procedure is key to achieve the desired data-dependent regret bound. One of the main technical features of OMD with log-barrier is its *restricted decision space*. Indeed, differently from most of the OMD-like algorithms, it is only allowed to select strategies belonging to the *truncated simplex*, which is defined as $\Omega := \left\{ \boldsymbol{x} \in \Delta_K : x(a) \geq \frac{1}{T} \ \forall a \in [K] \right\}$. This design choice avoids forced uniform exploration, and remarkably simplifies the analysis.

### 3.3  A Truncated Safe Decision Space

Our algorithms, presented in the following sections, rely on combining an OMD with log-barrier sub-routine with safe decision-making. This raises some challenges, since both components put some restrictions on the space from which strategies are chosen. In particular, OMD with log-barrier requires selecting strategies from the truncated simplex $\Omega$, while a safe decision-making involves choosing strategies from the safe decision space $\mathcal{S}_t$. Unfortunately, these two sets may in general be disjoint. Our algorithms implement a procedure that enables these two elements to work together. Specifically, at each round $t \in [T]$, they employ a larger safe decision space $\mathcal{S}_t^\circ$, which is defined as:

$$\mathcal{S}_t^\circ := \left\{ \boldsymbol{x} \in \Delta_K : (\widehat{\boldsymbol{g}}_{t,i} - \boldsymbol{\beta}_t)^\top \boldsymbol{x} \leq \alpha_i + \frac{K}{T} \ \forall i \in [m] \right\}.$$

This decision space is strictly larger than the safe decision space $\mathcal{S}_t$.

The following lemma characterizes the decision space obtained by intersecting $\Omega$ and $\mathcal{S}_t^\circ$: the *truncated safe decision space*.

**Lemma 2.** *For every $t \in [T]$, let $\widetilde{\mathcal{S}}_t := \Omega \cap \mathcal{S}_t^\circ$ be the intersection of the truncated simplex and the safe decision space. Then, under $\mathcal{E}(\delta)$, it holds that $\cap_{t \in [T]} \widetilde{\mathcal{S}}_t$ is non-empty.*

The above lemma can be proven by showing that any $\boldsymbol{x}^\circ \in \Delta_K$ that satisfies the constraints in expectation is included in $\mathcal{S}_t^\circ$. Moreover, $\|\boldsymbol{x}^\circ - \widetilde{\boldsymbol{x}}\|_1 \leq \frac{K}{T}$, where $\widetilde{\boldsymbol{x}} := \arg\min_{\boldsymbol{x} \in \Omega} \|\boldsymbol{x} - \boldsymbol{x}^\circ\|_1$. Thus, $\widetilde{\boldsymbol{x}}$ belongs to both $\Omega$ and $\mathcal{S}_t^\circ$. Additional details can be found in Appendix C. Intuitively, Lemma 2 states that, with high probability, at every round $t \in [T]$ there exists a strategy satisfying both the requirements of OMD with log-barrier and a "suitably-relaxed" safety condition.

## 4  Warm-Up: Small-Loss Guarantees in MABs with Soft Constraints

As a preliminary step toward our main result for the hard constraints settings, we design an algorithm with small-loss regret bound for *soft constraints* settings. Although these settings are technically easier than hard constraints ones, our algorithm incorporates some key components that are also central to the algorithm for hard constraints settings designed in Section 5. Moreover, the algorithm may also be of independent interest for other learning settings with soft constraints.

---

**Algorithm 1** `COLB`

---

**Require:** Learning rate $\eta > 0$, confidence $\delta \in (0,1)$, thresholds $\{\alpha_i\}_{i \in [m]}$

1: Define increase factor $\kappa \leftarrow e^{\frac{1}{\ln T}}$
2: Initialize $\boldsymbol{x}_1 \leftarrow \frac{1}{K}\mathbf{1}$, $h_{1,a} \leftarrow 2K$, $\eta_{1,a} \leftarrow \eta$ for all $a \in [K]$, $\widehat{\boldsymbol{g}}_{1,i} \leftarrow 0$ for all $i \in [m]$, $\boldsymbol{\beta}_1 \leftarrow \mathbf{1}$
3: **for** $t \in [T]$ **do**
4:      Select action $a_t \sim \boldsymbol{x}_t$
5:      Observe loss $\ell_t(a_t)$ and constraint costs $\boldsymbol{g}_{t,i}(a_t), \forall i \in [m]$

6:      Update $\widehat{\boldsymbol{g}}_{t,i}$ and $\boldsymbol{\beta}_t$ as described in Section 3.1
7:      Compute the safe decision space $\mathcal{S}_t^\circ$ as described in Section 3.3
8:      Compute the truncated safe decision space $\widetilde{\mathcal{S}}_t \leftarrow \Omega \cap \mathcal{S}_t^\circ$
9:      **if** $\widetilde{\mathcal{S}}_t$ is not empty **then**

10:          Compute $\widehat{\ell}_t(a) \leftarrow \frac{\ell_t(a)\mathbb{1}_{\{a_t=a\}}}{x_t(a)}, \forall a \in [K]$
11:          $\boldsymbol{x}_{t+1} \leftarrow \arg\min_{\boldsymbol{x} \in \widetilde{\mathcal{S}}_t} \widehat{\boldsymbol{\ell}}_t^\top \boldsymbol{x} + D_{\psi_t}(\boldsymbol{x}, \boldsymbol{x}_t)$, where $\psi_t(\boldsymbol{x}) = \sum_{a=1}^K \frac{1}{\eta_{t,a}} \ln \frac{1}{x(a)}$
12:          **for** $a \in [K]$ **do**
13:              **if** $\frac{1}{x_{t+1}(a)} > h_{t,a}$ **then**
14:                  $h_{t+1,a} \leftarrow \frac{2}{x_{t+1}(a)}$, $\eta_{t+1,a} \leftarrow \eta_{t,a}\kappa$
15:              **else**
16:                  $h_{t+1,a} \leftarrow h_{t,a}$, $\eta_{t+1,a} \leftarrow \eta_{t,a}$
17:      **else**
18:          Select strategy $\boldsymbol{x}_{t+1} \sim \Omega$ randomly

---

### 4.1 The `COLB` Algorithm

Algorithm 1 provides the pseudo-code of *Constrained OMD with Log-Barrier* (`COLB`). At a high level, the algorithm implements the combination of OMD with log-barrier and safe decision spaces introduced in Section 3. Indeed, Algorithm 1 is conceptually split into two main blocks. One block contains the set of instructions necessary to control constraint violations, highlighted in blue. The other block, highlighted in green, defines the OMD with log-barrier sub-routine. Algorithm 1 first defines a factor $\kappa$ that is employed to increase the learning rate $\eta_{t,a}$ for action $a$ if the probability of choosing the action falls below a certain threshold $h_{t,a}$ (Line 1). At each round, after playing an action and observing some feedback (Lines 4-5), the algorithm updates empirical means and confidence bounds for constraint costs (Line 6). Then, it builds the *truncated safe decision space* $\widetilde{\mathcal{S}}_t$ (Line 8), as described in Section 3. If this set is *not* empty (Line 9), then an update of OMD with log-barrier is performed over $\widetilde{\mathcal{S}}_t$ (Lines 10-11). Moreover, if the probability specified by computed strategy is too *small* for some action $a \in [K]$, *i.e.*, $1/x_{t+1}(a) \geq h_{t,a}$, then the learning rate $\eta_{t,a}$ is increased by a $\kappa$ factor and the threshold is increased to $2/x_{t+1}(a)$ (Line 13-16). Finally, if $\widetilde{\mathcal{S}}_t$ is empty, then a strategy is sampled randomly from the truncated simplex (Line 18).

In the following, we provide the theoretical guarantees attained by `COLB`, starting from constraint violations.

**Theorem 1.** *Let $\delta \in (0,1)$. Then, with probability at least $1 - 2\delta$, the* `COLB` *algorithm suffers cumulative positive constraint violations* $V_T \leq \mathcal{O}\left(\sqrt{KT \ln \left(TKm/\delta\right)}\right)$.

The proof of Theorem 1 relies on Lemma 2, and it can be found in Appendix C.3. The bound provided in the theorem matches, up to constant factors, the lower bound provided in Theorem 3 of [Bernasconi et al., 2022].

We are now ready to prove the main result of this section, which is a small-loss regret bound for `COLB`. This is stated in the following theorem.

**Theorem 2.** *Let $\delta \in (0,1)$ and $\eta = \min\left\{1/40H \ln T \ln(H/\delta), \sqrt{K/\sum_{t=1}^T \boldsymbol{\ell}_t^\top \boldsymbol{x}^* \ln(1/\delta)}\right\}$, where $H :=$ $\ln\left(\lceil\ln(T)\rceil\lceil 3\ln(T)\rceil/\delta\right)$. Then, with probability at least $1 - 4\delta$,* `COLB` *suffers a cumulative regret that can be bounded as* $R_T(\boldsymbol{\ell}_{1:T}) \leq \widetilde{\mathcal{O}}\left(\sqrt{K\sum_{t=1}^T \boldsymbol{\ell}_t^\top \boldsymbol{x}^* \ln(1/\delta)}\right)$.

Theorem 2 is proved by noticing that, even though the strategy update of Algorithm 1 works on changing decision spaces $\widetilde{\mathcal{S}}_t$, projecting does *not* prevent the OMD sub-routine from guaranteeing small-loss bounds. The proof can be found in Appendix C.2. Some remarks are in order.

**Remark 2** (Tightness). *The bound in Theorem 2 is tight, up to constants and logarithmic terms. Indeed, Theorem 3 of [Gerchinovitz and Lattimore, 2016] provides a regret lower bound of $\Omega(\sqrt{KL^*})$ in unconstrained MABs, where $L^*$ is the total loss of an optimal-in-hidsight action. The bound in Theorem 2 scales with the total loss of an optimal-in-hindsight randomized strategy $\boldsymbol{x}^*$. However, since in unconstrained settings an optimal (randomized) strategy is as powerful as an optimal action, the lower bound of [Gerchinovitz and Lattimore, 2016] carries over to our setting as well.*

**Remark 3** (Knowledge of $\sum_{t=1}^T \boldsymbol{\ell}_t^\top \boldsymbol{x}^*$). *Assuming knowledge of $L^*$ to set the learning rate is standard in the literature on small-loss bounds. As discussed in Remark 2 of [Allenberg et al., 2006] and Remark 1 of [Lee et al., 2020a], a doubling trick can relax this requirement, while ensuring that the regret bound does not deteriorate. This procedure is described in Appendix C.3 of [Lee et al., 2020b]. In our case, we require that $\sum_{t=1}^T \boldsymbol{\ell}_t^\top \boldsymbol{x}^*$ is known, instead of $L^*$. However the considerations made for $L^*$ still hold, and our algorithms can be made adaptive w.r.t. this quantity.*

# 5 Data-Dependent Guarantees in MABs with Hard Constraints

This section is entirely devoted to the main contribution of this paper, which is an algorithm that achieves a (tight) *data-dependent* regret bound in constrained MABs with *hard constraints*. The section begins by introducing the algorithm, which builds on the COLB algorithm introduced in Section 4 for soft constraints. After proving the guarantees attained by the algorithm, the section ends by proving a lower bound demonstrating that the regret bound of the algorithm is tight.

## 5.1 The SOLB Algorithm

Algorithm 2 provides the pseudo-code of *Safe OMD with log-barrier* (SOLB). Notice that the algorithm takes additional inputs compared to COLB. Specifically, it takes as inputs a *strictly feasible* strategy, *i.e.*, a strategy $\boldsymbol{x}^\diamond \in \Delta_K$ as defined in Assumption 2, and its associates costs $\{\theta_i\}_{i \in [m]}$.

Algorithm 2 highlights in pink its differences with respect to Algorithm 1. The key difference between COLB and SOLB is that the strategy chosen by the latter is *not* readily the one selected through the OMD update. Specifically, at each round $t$, SOLB plays a convex combination between the strictly feasible strategy $\boldsymbol{x}^\diamond$ given as input and the one selected by OMD, denoted $\widetilde{\boldsymbol{x}}_t$. The combination factor $\gamma_t$ is chosen in an adaptive way to guarantee that the resulting strategy $\boldsymbol{x}_t$ satisfies the constraints with high probability. Intuitively, the combination factor $\gamma_t$ weights how safe is to play the strategy computed by OMD rather than the strictly feasible strategy. If the strategy $\widetilde{\boldsymbol{x}}_t$ produced by the OMD update satisfies the constraints with high probability, then $\gamma_t$ is set to zero, and the algorithm selects strategy $\widetilde{\boldsymbol{x}}_t$. A larger $\gamma_t$ weights more the strictly feasible strategy $\boldsymbol{x}^\diamond$ than $\widetilde{\boldsymbol{x}}_t$. In the first rounds, since confidence intervals for cost estimates are large, $\gamma_t$ is strictly greater than zero, while, as the confidence intervals become smaller, $\gamma_t$ approaches zero. As in Algorithm 1, when the truncated safe decision space is *not* empty, the algorithm runs a step of the OMD with log-barrier sub-routine. Then, a new combination factor is computed (Line 19). On the other hand, if the truncated safe decision space is empty, then $\gamma_t$ is set to one (Line 23), and the algorithm thus selects the known strictly feasible strategy. In Figure 1 in Appendix A, we provide a graphical intuition on how $\boldsymbol{x}_t$ is selected when $K = 3$.

## 5.2 Theoretical Guarantees of SOLB

In this section, we provide the theoretical guarantees attained by SOLB. We start by showing that SOLB satisfies the constraints at every round with high probability.

**Theorem 3.** *Let $\delta \in (0, 1)$. With probability at least $1 - \delta$, SOLB guarantees that $\boldsymbol{g}_i^\top \boldsymbol{x}_t \leq \alpha_i$ holds for every constraint $i \in [m]$ and round $t \in [T]$.*

Theorem 3 can be proven by analyzing the behavior of the combination factor $\gamma_t$. We show that $\gamma_t$ is large enough to compensate the violations potentially suffered by the strategy $\widetilde{\boldsymbol{x}}_{t+1}$ computed by the OMD update. Specifically, $\gamma_t = 0$ when $\widetilde{\boldsymbol{x}}_{t+1}$ satisfies the constraints with high probability.

**Algorithm 2** SOLB

---

**Require:** Learning rate $\eta > 0$, confidence parameter $\delta \in (0,1)$, thresholds $\{\alpha_i\}_{i \in [m]}$, *strictly* feasible strategy $\boldsymbol{x}^\diamond \in \Delta_K$ with its associated costs $\{\theta_i\}_{i \in [m]}$ (see Assumption 2)

1: Define increase factor $\kappa \leftarrow e^{\frac{1}{\ln T}}$
2: Initialize $\widetilde{\boldsymbol{x}}_1 \leftarrow \frac{1}{K}\mathbf{1}$, $\rho_{1,a} \leftarrow 2K$, $\eta_{1,a} \leftarrow \eta$ for all $a \in [K]$, $\widehat{\boldsymbol{g}}_{1,i} \leftarrow 0$ for all $i \in [m]$, $\boldsymbol{\beta}_1 \leftarrow \mathbf{1}$
3: Initialize $\gamma_0 \leftarrow \max_{i \in [m]} \frac{1-\alpha_i}{1-\theta_i}$
4: Select $\boldsymbol{x}_1 \leftarrow \gamma_0 \boldsymbol{x}^\diamond + (1-\gamma_0)\widetilde{\boldsymbol{x}}_1$
5: **for** $t \in [T]$ **do**
6:     Select action $a_t \sim \boldsymbol{x}_t$
7:     Observe loss $\ell_t(a_t)$ and constraint costs $\boldsymbol{g}_{t,i}(a_t)$, $\forall i \in [m]$
8:     Update $\widehat{\boldsymbol{g}}_{t,i}$ and $\boldsymbol{\beta}_t$ as described in Section 3.1
9:     Compute the safe decision space $\mathcal{S}_t^\circ$ as described in Section 3.3
10:     Compute the truncated safe decision space $\widetilde{\mathcal{S}}_t \leftarrow \Omega \cap \mathcal{S}_t^\circ$
11:     **if** $\widetilde{\mathcal{S}}_t$ is not empty **then**
12:         Compute $\widehat{\ell}_t(a) \leftarrow \frac{\ell_t(a)\mathbb{1}_{\{a_t = a\}}}{x_t(a)}$, $\forall a \in [K]$
13:         $\widetilde{\boldsymbol{x}}_{t+1} \leftarrow \arg\min_{\boldsymbol{x} \in \widetilde{\mathcal{S}}_t} \widehat{\boldsymbol{\ell}}_t^\top \boldsymbol{x} + D_{\psi_t}(\boldsymbol{x}, \boldsymbol{x}_t)$, where $\psi_t(\boldsymbol{x}) = \sum_{a=1}^K \frac{1}{\eta_{t,a}} \ln \frac{1}{x(a)}$
14:         **for** $a \in [K]$ **do**
15:             **if** $\frac{1}{x_{t+1}(a)} > h_{t,a}$ **then**
16:                 $h_{t+1,a} \leftarrow \frac{2}{x_{t+1}(a)}$, $\eta_{t+1,a} \leftarrow \eta_{t,a}\kappa$
17:             **else**
18:                 $h_{t+1,a} \leftarrow h_{t,a}$, $\eta_{t+1,a} \leftarrow \eta_{t,a}$
19:         Compute the combination factor:

$$\gamma_t \leftarrow \begin{cases} \max_{i \in \mathcal{M}} \left\{ \frac{\min\{(\widehat{\boldsymbol{g}}_{t,i} + \boldsymbol{\beta}_t)^\top \widetilde{\boldsymbol{x}}_{t+1}, 1\} - \alpha_i}{\min\{(\widehat{\boldsymbol{g}}_{t,i} + \boldsymbol{\beta}_t)^\top \widetilde{\boldsymbol{x}}_{t+1}, 1\} - \theta_i} \right\} & \text{if } \mathcal{E} \text{ holds} \\ 0 & \text{otherwise} \end{cases},$$

20:         where $\mathcal{M} := \{i \in [m] : (\widehat{\boldsymbol{g}}_{t,i} + \boldsymbol{\beta}_t)^\top \widetilde{\boldsymbol{x}}_{t+1} > \alpha_i\}$, $\mathcal{E} = \{\exists i \in [m] : (\widehat{\boldsymbol{g}}_{t,i} + \boldsymbol{\beta}_t)^\top \widetilde{\boldsymbol{x}}_{t+1} > \alpha_i\}$
21:     **else**
22:         Select strategy $\widetilde{\boldsymbol{x}}_{t+1} \sim \Omega$ randomly
23:         Set the combination factor to $\gamma_t \leftarrow 1$
24:     $\boldsymbol{x}_{t+1} \leftarrow \gamma_t \boldsymbol{x}^\diamond + (1-\gamma_t)\widetilde{\boldsymbol{x}}_{t+1}$

---

Otherwise, $\gamma_t$ is proportional to the pessimistic violation that $\widetilde{\boldsymbol{x}}_{t+1}$ would suffer. Assuming that $\boldsymbol{x}^\diamond$ is strictly feasible implies that $\gamma_t < 1$ in every round $t$ in which the truncated safe decision space is non-empty. To see this, notice that $\gamma_t \leq \max_{i \in [m]} \frac{1-\alpha_i}{1-\theta_i} \leq \max_{i \in [m]} \frac{1-\alpha_i}{1-\alpha_i+\rho} < 1$. Thus, a minimum amount of exploration is always guaranteed. When the truncated safe decision space is empty, the algorithm uses the strictly feasible strategy.

We now analyze the regret suffered by SOLB. Before presenting our main result, we introduce some technical lemmas that are useful to understand the nature of the regret bound.

**Lemma 3.** *Let* $\delta \in (0,1)$ *and* $\rho \geq \frac{12K}{T}$. *Then, with probability at least* $1 - 2\delta$, SOLB *satisfies:*

$$R_T^\diamond(\boldsymbol{\ell}_{1:T}) \leq \mathcal{O}\left( \frac{K}{\rho} \sqrt{\sum_{t=1}^T \left(\boldsymbol{\ell}_t^\top(\boldsymbol{x}^\diamond - \boldsymbol{x}^*)\right)^2 \cdot \ln\left(\frac{KTm}{\delta}\right)} + \frac{K}{\rho^6} \ln\left(\frac{KTm}{\delta}\right) \right), \qquad (2)$$

*where* $R_T^\diamond(\boldsymbol{\ell}_{1:T}) := \sum_{t=1}^T \gamma_{t-1} \boldsymbol{\ell}_t^\top(\boldsymbol{x}^\diamond - \boldsymbol{x}^*)$.

Intuitively, the above lemma states that the regret accrued by the strictly feasible strategy, when weighted by the combination factors, is not *too large*. Specifically, the term that appears in Equation (2) is the scaled Euclidean norm of the sequence of instantaneous regrets suffered by $\boldsymbol{x}^\diamond$. This quantity represents some sort of distance between an optimal strategy and the strictly feasible one. At the end of this section, we provide a discussion on the role of this quantity, showing that it represents a source of complexity for the problem instance.

**Lemma 4.** *Let $\delta \in (0,1)$ and $\eta \leq \frac{\rho}{40 H \ln T \ln(H/\delta)}$, where $H := \ln\left(\lceil \ln(T) \rceil \lceil 3\ln(T) \rceil / \delta\right)$. Then, with probability at least $1 - 2\delta$, SOLB satisfies:*

$$\widetilde{R}_T(\boldsymbol{\ell}_{1:T}) \leq \mathcal{O}\left( \frac{K}{\eta} + \frac{\eta}{\rho}\sum_{t=1}^T \boldsymbol{\ell}_t^\top \boldsymbol{x}_t + \frac{K\eta}{\rho}\ln\left(\frac{1}{\delta}\right) + \sqrt{\sum_{t=1}^T \boldsymbol{\ell}_t^\top \boldsymbol{x}_t \ln\left(\frac{1}{\delta}\right)} + \frac{\eta}{\rho}\sum_{t=1}^T \boldsymbol{\ell}_t^\top \boldsymbol{x}^* \ln\left(\frac{1}{\delta}\right) \right),$$

*where $\widetilde{R}_T(\boldsymbol{\ell}_{1:T}) = \sum_{t=1}^T (1 - \gamma_{t-1})\boldsymbol{\ell}_t^\top (\widetilde{\boldsymbol{x}}_t - \boldsymbol{x}^*)$.*

Lemma 4 provides an upper bound on the regret accrued by the strategy proposed by OMD, when weighted by $1 - \gamma_t$. The bound seems involved. However, the terms we are more interested in are the second and the fifth, which dominate the other ones. In particular, we highlight how the quantity $\widetilde{R}_T(\boldsymbol{\ell}_{1:T})$ is bounded by a sum of the total loss of the strategy played by SOLB and the total loss of the optimal strategy. This provides a link between the total loss of the strategy proposed by OMD, the total loss of the strategy that is actually played by SOLB, and the total loss of the optimal strategy. Keeping in mind that our goal is to obtain a regret bound for SOLB that scales with the latter, Lemma 4 plays a key role in the final result. The proofs of the lemmas are in Appendix D.1.

We are now ready to present our main result, *i.e.*, a high-probability regret bound for SOLB.

**Theorem 4.** *Let $\delta \in (0,1)$, $\rho \geq \frac{12K}{T}$, and $\eta = \min\left\{ \frac{\rho}{40 H \ln T \ln(H/\delta)}, \sqrt{K / \sum_{t=1}^T \boldsymbol{\ell}_t^\top \boldsymbol{x}^* \ln(1/\delta)} \right\}$, where $H := \ln\left(\lceil \ln(T) \rceil \lceil 3\ln(T) \rceil / \delta\right)$. Then, SOLB suffers a cumulative regret bounded as:*

$$R_T(\boldsymbol{\ell}_{1:T}) \leq \widetilde{\mathcal{O}}\Bigg( \underbrace{\frac{K \ln(1/\delta)}{\rho}\sqrt{\sum_{t=1}^T \left(\boldsymbol{\ell}_t^\top (\boldsymbol{x}^\diamond - \boldsymbol{x}^*)\right)^2}}_{\text{(A) Safety Complexity}} + \underbrace{\frac{1}{\rho}\sqrt{K \sum_{t=1}^T \boldsymbol{\ell}_t^\top \boldsymbol{x}^* \ln\left(\frac{1}{\delta}\right)}}_{\text{(B) Bandit Complexity}} \Bigg), \qquad (3)$$

*where $\widetilde{\mathcal{O}}$ hides universal constants and logarithmic terms not depending on $\delta$.*

Theorem 4 is proved by decomposing the regret in the quantities analyzed in Lemmas 3 and 4. By construction, $\boldsymbol{x}_t = \gamma_{t-1}\boldsymbol{x}^\diamond + (1 - \gamma_{t-1})\widetilde{\boldsymbol{x}}_t$, which implies $R_T(\boldsymbol{\ell}_{1:T}) = R_T^\diamond(\boldsymbol{\ell}_{1:T}) + \widetilde{R}_T(\boldsymbol{\ell}_{1:T})$. Thus, we can sum the upper bounds presented in Lemmas 3 and 4. Finally, noting that, by definition of $\eta$, it holds $\frac{\eta}{\rho}\sum_{t=1}^T \boldsymbol{\ell}_t^\top \boldsymbol{x}_t \leq \frac{1}{2}R_T + \frac{\eta}{\rho}\sum_{t=1}^T \boldsymbol{\ell}_t^\top \boldsymbol{x}^*$, we can solve the deriving quadratic inequality in $R_T$, which yields Equation (3). A detailed proof of Theorem 4 is in Appendix D.1. Theorem 4 is one of the main results of this paper. It establishes a regret bound that depends on two contributions: the total loss incurred by an optimal-in-hindsight strategy (B), and the total squared difference between the losses of the strictly feasible strategy and the benchmark (A). This result provides a natural interpretation on the intrinsic difficulty of MABs with hard constraints. On the one hand, our bound scales as the performance of an optimal strategy, which is common to any small-loss bound in unconstrained MABs. We call this contribution Bandit Complexity, as it represents the complexity of learning independently of the presence of the constraints. On the other hand, we pay an additional term—peculiar of our setting—that encodes the distance between the benchmark and the strictly feasible strategy given as input to the algorithm. We call this contribution Safety Complexity, as it represents the complexity of learning an optimal feasible strategy while satisfying the constraints at every round with high probability. We acknowledge that the state-of-the-art bounds in online settings with hard constraints are of order $\widetilde{\mathcal{O}}(1/\rho \sqrt{T})$ (*e.g.*, [Pacchiano et al., 2021]). Indeed, our bound not only improves the aforementioned result in the best case, while being equivalent in the worst one, but it also decomposes the former in two quantities that are easily interpretable.

## 5.3 A Small-Loss Style Regret Lower Bound

In this section, we provide a small-loss style regret lower bound for the hard constrained bandit problem. In [Gerchinovitz and Lattimore, 2016] the authors show that, in adversarial non-constrained bandit problems, the regret suffered by every algorithm is lower bounded as $\mathcal{O}(\sqrt{L^*})$, where $L^*$ is the total loss accrued by the benchmark. This term is represented, in our setting, by the Bandit Complexity contribution. In our setting, the strictly feasible strategy $\boldsymbol{x}^\diamond$ is crucial in defining how hard is an instance: in fact, Equation (3) shows that SOLB benefits instances where the optimum is *close* (in terms of performances) to the strictly feasible strategy, which translates in a small Safety Complexity. This

behavior is natural as the strictly feasible strategy represents a starting point for the exploration, and the algorithm remains somehow tied to that. This raises the natural question on whether this double dependency, one on the optimal total loss and the other on the difference with the strictly feasible strategy, is actually tight. The next two results bridge Theorem 4 with the standard literature results, *i.e.* regret bounds depending on $T$, and show that the performance of SOLB is optimal when disregarding logarithmic terms. We start by introducing an important technical notion, that is handy in bridging constrained small-loss bounds and standard bounds depending on $T$ only. We define the *constrained small-loss balls* as $\mathcal{B}_{\omega,\Delta,T} := \left\{ \boldsymbol{\ell}_{1:T} \in [0,1]^{KT} : \frac{\sum_{t=1}^{T} \boldsymbol{\ell}_t^\top \boldsymbol{x}^*}{T} \leq \omega \cap \frac{\sum_{t=1}^{T} (\boldsymbol{\ell}_t^\top (\boldsymbol{x}^\diamond - \boldsymbol{x}^*))^2}{T} \leq \Delta^2 \right\}$. The quantities $\omega, \Delta \in [0,1]$ represent two different sources of difficulty for the learning algorithm: $\omega$ express how difficult is the identification of the optimum as in a non-constrained bandit problem, and $\Delta$ represents the additional difficulty provided by the constraints satisfaction. The next result allows us to rephrase the regret upper bound provided in Theorem 4 in terms of the constrained small-loss ball. In fact, it is a trivial consequence of Equation (3) and the definition of constrained small-loss ball.

**Corollary 1.** *For all* $\omega \in [0,1]$ *and for all* $\Delta \in [0,1]$*, it holds* $\sup_{\boldsymbol{\ell}_{1:T} \in \mathcal{B}_{\omega,\Delta,T}} \mathbb{E}[R_T(\boldsymbol{\ell}_{1:T})] \leq \widetilde{\mathcal{O}} \left( \frac{K\Delta}{\rho} \sqrt{T} + \frac{1}{\rho} \sqrt{K\omega T} \right),$ *where the expectation is taken w.r.t. the internal algorithm randomization.*

Notice that $\omega$ and $\Delta$ can be treated as instance-dependent parameters, as they represent how far an instance is from the worst-case one, *i.e.* when those two are both equal to $1$. Finally, the next result shows that no algorithm can have a better dependence on the parameters $\omega$ and $\Delta$.

**Theorem 5.** *Let* $K \geq 2$*,* $T \geq \max \left\{ 2, (11 + \ln T) \left( \frac{8}{3} \right)^2 \right\}$*, and* $\omega \in \left[ \frac{1}{T} \left( \frac{11}{2} + \ln T \right), \frac{1}{2} \right]$*. Then for every randomized algorithm, we have* $\sup_{\boldsymbol{\ell}_{1:T} \in \mathcal{B}_{\omega,\Delta,T}} \mathbb{E}[R_T(\boldsymbol{\ell}_{1:T})] \geq \Omega \left( \frac{\Delta}{\rho} \sqrt{T} + \sqrt{\omega T} \right),$ *where the expectation is taken with respect to the internal randomization of the algorithm.*

A detailed proof of Theorem 5 can be found in Appendix D.3. This result shows that SOLB achieves an optimal dependence on both $\Delta$ and $\omega$.

Finally, we leave as an open question whether the dependence on the constant $\rho$ in contribution (B) of the regret upper bound can be removed. Notice that $\rho$ is a constant tied to the strictly feasible strategy and it encompasses the difficulty provided by the constraints to the instance. Thus, we believe that the $^1/_\rho$ dependence should affect contribution (A) only and we conjecture that our lower bound is actually tight, while it is the upper bound that can be lowered.

## Acknowledgments

This paper is supported by the Italian MIUR PRIN 2022 Project "Targeted Learning Dynamics: Computing Efficient and Fair Equilibria through No-Regret Algorithms", by the FAIR (Future Artificial Intelligence Research) project, funded by the NextGenerationEU program within the PNRR-PE-AI scheme (M4C2, Investment 1.3, Line on Artificial Intelligence), and by the EU Horizon project ELIAS (European Lighthouse of AI for Sustainability, No. 101120237).

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

# Appendix

The Appendix is structured as follows:

- In Appendix A, we provide a graphical representation of Algorithm 2's update.
- In Appendix B, we provide the complete discussion on related works.
- In Appendix C, we provide the omitted analysis for the *soft constraints* setting.
- In Appendix D, we provide the omitted analysis for the *hard constraints* setting.

## A   Graphical Representation of the Update of Algorithm 2

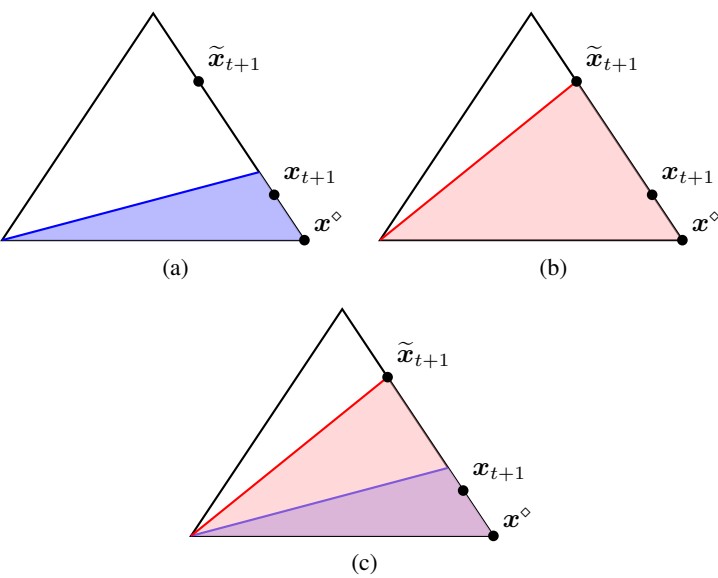

Figure 1: A graphical representation of the update performed by Algorithm 2. For the sake of exposition, we omit the constraint $x(a) \geq 1/T, \forall a \in [K]$. Specifically, in Figure 1(a), we provide the graphical representation of the safe subset of the simplex. Notice that, the strictly safe strategy $\boldsymbol{x}^\diamond$ associated to the Slater's parameter $\rho$ is a vertex of the simplex. In Figure 1(b), we provide the graphical representation of the *estimated* safe decision space. Notice that, $\widetilde{\boldsymbol{x}}_{t+1}$ lies on this set as prescribed by the projection of both Algorithm 1 and Algorithm 2. Finally, in Figure 1(c), we provide how the convex combination is performed. Notice that, $\boldsymbol{x}_{t+1}$ is computed pessimistically. Indeed, due to the high uncertainty in the constraints estimation, $\boldsymbol{x}_{t+1}$ is an interior point of the safe space.

## B   Related Works

**Data-Dependent Regret Bounds**   Over the last two decades, in the literature on (unconstrained) adversarial MABs there has been an increasing interest in providing regret guarantees that depend on the difficulty of the specific instance faced by algorithm. Such bounds are usually referred to as *data-dependent bounds* or *instance-dependent bounds*. Examples include, but are not limited to, *small-loss bounds* (also called *first-order bounds*). An algorithm enjoying small-loss guarantees has its regret upper bound scaling with $\mathcal{O}(\sqrt{L^*})$, where $L^* \leq T$ is the total loss accrued by the benchmark. Small loss bounds over the *expected* regret have been obtained in several settings, from MABs to contextual bandits [Allenberg et al., 2006, Neu, 2015b, Allen-Zhu et al., 2018, Lee et al., 2020b]. However, few works managed to recover small-loss guarantees that hold with high probability [Neu, 2015a, Lee et al., 2020a]. The tightness of small-loss bounds has been analyzed only recently. In particular, in [Gerchinovitz and Lattimore, 2016] the authors provide data-dependent lower bounds for adversarial MABs, showing that the $\mathcal{O}(\sqrt{L^*})$ rate cannot be improved.

**Constrained Online Learning** Online leaning with *unknown* constraints has been widely investigated (see, *e.g.*, [Mannor et al., 2009, Liakopoulos et al., 2019, Castiglioni et al., 2022a,b]). Two main settings are usually studied. In *soft constraints* settings (see, *e.g.*, [Chen et al., 2022]), the aim is to guarantee that the constraint violations incurred by the algorithm grow sub-linearly. In *hard constraints* settings, the algorithms must satisfy the constraints at every round, by assuming knowledge of a strictly feasible decision (see, *e.g.*, [Pacchiano et al., 2021]). Both soft and hard constraints have been generalized to settings that are more challenging than MABs, such as linear bandits (see, *e.g.*, [Gangrade et al., 2024]) and games (see, *e.g.*, [Bernasconi et al., 2022]). None of these works in the constrained online learning literature studies data-dependent regret bounds. There also exists a huge literature on constrained Markov decision processes (see, *e.g.*, [Wei et al., 2018, Zheng and Ratliff, 2020, Bai et al., 2020, Efroni et al., 2020, Qiu et al., 2020, Ding et al., 2021, Liu et al., 2021, Wei et al., 2022, 2023, Ding and Lavaei, 2023, Stradi et al., 2024b, Müller et al., 2024, Stradi et al., 2025, 2024a]). Most of these works focus on stochastic settings, and none of them provides data-dependent regret bounds. Finally, some works focus on constrained online convex optimization settings (see, *e.g.*, [Mahdavi et al., 2012, Jenatton et al., 2016, Yu et al., 2017]). Nonetheless, they do not provide data-dependent regret bounds.

# C Omitted Proofs for Soft Constraints

## C.1 Non-Emptiness of the Truncated Safe Decision Space

**Lemma 2.** *For every $t \in [T]$, let $\widetilde{\mathcal{S}}_t := \Omega \cap \mathcal{S}_t^\circ$ be the intersection of the truncated simplex and the safe decision space. Then, under $\mathcal{E}(\delta)$, it holds that $\cap_{t \in [T]} \widetilde{\mathcal{S}}_t$ is non-empty.*

*Proof.* Consider a feasible strategy $\boldsymbol{x}^\circ$, namely, $\boldsymbol{g}_i^\top \boldsymbol{x}^\circ \leq \alpha_i$ for all $i \in [m]$. Under the event $\mathcal{E}(\delta)$, Lemma 1 implies

$$\left(\widehat{\boldsymbol{g}}_{t,i} - \boldsymbol{\beta}_t\right)^\top \boldsymbol{x}^\circ \leq \alpha_i, \quad \forall t \in [T]. \tag{4}$$

Notice that, since $\boldsymbol{x}^*$ is feasible, the aforementioned reasoning still holds for the constrained optimal solution.

Then, notice that for any $\boldsymbol{x} \in \Delta_K$, there exists a strategy $\widetilde{\boldsymbol{x}} \in \Omega$ s.t. $\|\widetilde{\boldsymbol{x}} - \boldsymbol{x}\|_1 \leq \frac{K}{T}$.

Thus, employing the reasoning above and taking:

$$\widetilde{\boldsymbol{x}} := \arg\min_{\boldsymbol{x} \in \Omega} \|\boldsymbol{x} - \boldsymbol{x}^\circ\|_1,$$

we have, for all $i \in [m]$ and for all $t \in [T]$:

$$\begin{aligned}
(\widehat{\boldsymbol{g}}_{t,i} - \boldsymbol{\beta}_t)^\top \widetilde{\boldsymbol{x}} &= (\widehat{\boldsymbol{g}}_{t,i} - \boldsymbol{\beta}_t)^\top (\widetilde{\boldsymbol{x}} \pm \boldsymbol{x}^\circ) \\
&= (\widehat{\boldsymbol{g}}_{t,i} - \boldsymbol{\beta}_t)^\top \boldsymbol{x}^\circ + (\widehat{\boldsymbol{g}}_{t,i} - \boldsymbol{\beta}_t)^\top (\widetilde{\boldsymbol{x}} - \boldsymbol{x}^\circ) \\
&\leq \alpha_i + \|\widetilde{\boldsymbol{x}} - \boldsymbol{x}^\circ\|_1 \\
&\leq \alpha_i + \frac{K}{T},
\end{aligned}$$

which holds with probability at least $1 - \delta$ and implies that $\widetilde{\mathcal{S}}_t$ is never empty.

To conclude the proof, we notice that:

$$\begin{aligned}
\bigcap_{t \in [T]} \widetilde{\mathcal{S}}_t &= \bigcap_{t \in [T]} (\Omega \cap \mathcal{S}_t^\circ) \\
&= \Omega \cap \left(\bigcap_{t \in [T]} \mathcal{S}_t^\circ\right).
\end{aligned}$$

Noticing that, by Equation (4) and employing the same reasoning above, it holds:

$$\widetilde{\boldsymbol{x}} \in \bigcap_{t \in [T]} \widetilde{\mathcal{S}}_t,$$

concludes the proof. $\qquad\square$

### C.2 Cumulative Regret of COLB

**Lemma 5.** *For any $\delta \in (0,1)$, for any $\boldsymbol{u}$ s.t. $\boldsymbol{u} \in \cap_{t \in [T]} \widetilde{\mathcal{S}}_t$, Algorithm 1 guarantees the following bound:*

$$\sum_{t=1}^{T} \widehat{\boldsymbol{\ell}}_t^{\top} (\boldsymbol{x}_t - \boldsymbol{u}) \leq \mathcal{O}\left(\frac{K \ln T}{\eta} + \eta \sum_{t=1}^{T} \ell_t(a_t)\right) - \frac{\boldsymbol{h}_T^{\top} \boldsymbol{u}}{10 \eta \ln T},$$

*with probability at least $1 - \delta$.*

*Proof.* We fix $\boldsymbol{u}$ s.t. $\boldsymbol{u} \in \cap_{t \in [T]} \widetilde{\mathcal{S}}_t$, that is, any possible vector belonging to the intersection between the safe sets built by Algorithm 1. Then, we apply standard OMD with log-barrier results (see [Agarwal et al., 2017]) to obtain:

$$\sum_{t=1}^{T} \widehat{\boldsymbol{\ell}}_t^{\top} (\boldsymbol{x}_t - \boldsymbol{u}) \leq \sum_{t=1}^{T} \left(D_{\psi_t}(\boldsymbol{u}, \boldsymbol{x}_t) - D_{\psi_t}(\boldsymbol{u}, \boldsymbol{x}_{t+1})\right) + \sum_{t=1}^{T} \sum_{a=1}^{K} \eta_{t,a} x_t^2(a) \widehat{\ell}_t^2(a). \quad (5)$$

The result above holds since $\widetilde{\mathcal{S}}_t$ is a polytope (thus, convex) for any $t \in [T]$. As $\widetilde{\mathcal{S}}_t$ is included in $\Delta_K$, then the intersection is convex. Moreover, we notice that, by Lemma 2, we have, with probability at least $1 - \delta$, that $\cap_{t \in [T]} \widetilde{\mathcal{S}}_t$ is non-empty, by construction of the estimated safe set. Equation (5) holds under the event mentioned above.

For the last term of Equation (5), we simply notice that, for any $t \in [T]$, it holds:

$$\eta_{t,a} x_t^2(a) \widehat{\ell}_t^2(a) = \eta_{t,a} x_t^2(a) \frac{\ell_t^2(a)}{x_t^2(a)} \mathbb{1}\{a_t = a\}$$

$$\leq \eta_{t,a_t} \ell_t^2(a_t)$$

$$\leq \eta_{t,a_t} \ell_t(a_t)$$

$$\leq \eta_{T,a_t} \ell_t(a_t)$$

$$\leq 5\eta \ell_t(a_t),$$

where the last inequality holds since $\eta_{T,a} = \kappa^{n_a} \eta_{1,a}$, where $n_a$ is the number of times Algorithm 1 increases the learning rate for arm $a$, and $\kappa^{n_a} \leq 5$.

In the following, we define $h(y) = y - 1 - \ln y$. Thus we bound the first two term of Equation (5) as follows:

$$\sum_{t=1}^{T} \left(D_{\psi_t}(\boldsymbol{u}, \boldsymbol{x}_t) - D_{\psi_t}(\boldsymbol{u}, \boldsymbol{x}_{t+1})\right)$$

$$\leq D_{\psi_1}(\boldsymbol{u}, \boldsymbol{x}_1) + \sum_{t=1}^{T-1} \left(D_{\psi_{t+1}}(\boldsymbol{u}, \boldsymbol{x}_{t+1}) - D_{\psi_t}(\boldsymbol{u}, \boldsymbol{x}_{t+1})\right) \quad (6)$$

$$= \frac{1}{\eta} \sum_{a=1}^{K} h\left(\frac{u(a)}{x_1(a)}\right) + \sum_{a=1}^{K} \sum_{t=1}^{T-1} \left(\frac{1}{\eta_{t+1,a}} - \frac{1}{\eta_{t,a}}\right) h\left(\frac{u(a)}{x_{t+1}(a)}\right),$$

where Inequality (6) holds since the Bregman is always greater or equal than zero.

Thus, we focus on the first term, bounding it as follows:

$$\frac{1}{\eta} \sum_{a=1}^{K} h\left(\frac{u(a)}{x_1(a)}\right) = \frac{1}{\eta} \sum_{a=1}^{K} - \ln(K u(a)) \quad (7)$$

$$\leq \frac{K \ln T}{\eta}, \quad (8)$$

where Equation (7) holds since $\boldsymbol{x}_t$ is initialized uniformly and Inequality (8) holds since $u(a) \geq 1/T$ for all $a \in [K]$.

To bound the final term, we will refer as $t_a$ to the last time step where the learning rate of arm $a$ is increased. Thus, we proceed as follows.

$$\left(\frac{1}{\eta_{t_j+1,a}} - \frac{1}{\eta_{t_j,a}}\right) h\left(\frac{u(a)}{x_{t_j+1}(a)}\right) = \frac{1 - \kappa}{k^{n_j} \eta} h\left(\frac{u(a)}{x_{t_j+1}(a)}\right)$$

$$\leq \frac{-h\left(\frac{u(a)}{x_{t_j+1}(a)}\right)}{5\eta\ln T}$$

$$= \frac{-h\left(\frac{u(a)h_{T,a}}{2}\right)}{5\eta\ln T}$$

$$= \frac{\ln\left(\frac{u(a)h_{T,a}}{2}\right) - \frac{u(a)h_{T,a}}{2} + 1}{5\eta\ln T}$$

$$\leq \frac{\ln T - \frac{u(a)h_{T,a}}{2} + 1}{5\eta\ln T},$$

where we used that $1 - \kappa \leq -\frac{1}{\ln T}$ and $\frac{u(a)h_{T,a}}{2} \leq \frac{1}{x_{t_j+1}(a)} \leq T$.

Combining the previous bounds concludes the proof. $\qquad\square$

**Theorem 2.** *Let $\delta \in (0,1)$ and $\eta = \min\left\{\frac{1}{40H\ln T\ln(H/\delta)}, \sqrt{K/\sum_{t=1}^{T}\boldsymbol{\ell}_t^\top\boldsymbol{x}^*\ln(1/\delta)}\right\}$, where $H :=$ $\ln\left(\lceil\ln(T)\rceil\lceil 3\ln(T)\rceil/\delta\right)$. Then, with probability at least $1 - 4\delta$,* COLB *suffers a cumulative regret that can be bounded as $R_T(\boldsymbol{\ell}_{1:T}) \leq \tilde{\mathcal{O}}\left(\sqrt{K\sum_{t=1}^{T}\boldsymbol{\ell}_t^\top\boldsymbol{x}^*\ln(1/\delta)}\right)$.*

*Proof.* We first decompose the regret as follows:

$$R_T := \sum_{t=1}^{T}\boldsymbol{\ell}_t^\top(\boldsymbol{x}_t - \boldsymbol{x}^*)$$

$$= \sum_{t=1}^{T}\widehat{\boldsymbol{\ell}}_t^\top(\boldsymbol{x}_t - \boldsymbol{u}) + \sum_{t=1}^{T}(\boldsymbol{\ell}_t - \widehat{\boldsymbol{\ell}}_t)^\top\boldsymbol{x}_t + \sum_{t=1}^{T}(\widehat{\boldsymbol{\ell}}_t - \boldsymbol{\ell}_t)^\top\boldsymbol{u} + \sum_{t=1}^{T}\boldsymbol{\ell}_t^\top(\boldsymbol{u} - \boldsymbol{x}^*)$$

$$\leq \sum_{t=1}^{T}\widehat{\boldsymbol{\ell}}_t^\top(\boldsymbol{x}_t - \boldsymbol{u}) + \sum_{t=1}^{T}(\boldsymbol{\ell}_t - \widehat{\boldsymbol{\ell}}_t)^\top\boldsymbol{x}_t + \sum_{t=1}^{T}(\widehat{\boldsymbol{\ell}}_t - \boldsymbol{\ell}_t)^\top\boldsymbol{u} + K,$$

where we take $\boldsymbol{u}$ as $\min_{\boldsymbol{u}\in\cap_{t\in[T]}\widetilde{\mathcal{S}}_t}\|\boldsymbol{x}^* - \boldsymbol{u}\|_1$ and the inequality follows from the Hölder inequality after noticing that, under the event of Lemma 2, which holds with probability at least $1 - \delta$, the maximum $\ell_1$ distance between the safe optimum $\boldsymbol{x}^*$ and $\boldsymbol{u}$ is $K/T$.

We will bound the remaining quantities separately.

**Bound on the first term** The first term follows from Lemma 5.

**Bound on the second term** To bound the second term we notice that it is a Martingale difference sequence, where any difference is bounded as:

$$\left|\left(\mathbb{E}_t\left[\widehat{\boldsymbol{\ell}}_t\right] - \widehat{\boldsymbol{\ell}}_t\right)^\top\boldsymbol{x}_t\right| \leq \max\left\{\widehat{\boldsymbol{\ell}}_t^\top\boldsymbol{x}_t, \mathbb{E}_t\left[\widehat{\boldsymbol{\ell}}_t\right]^\top\boldsymbol{x}_t\right\}$$

$$= \max\left\{\sum_{a=1}^{K}x_t(a)\frac{\ell_t(a)}{x_t(a)}\mathbb{1}\{a_t = a\}, \mathbb{E}_t\left[\widehat{\boldsymbol{\ell}}_t\right]^\top\boldsymbol{x}_t\right\}$$

$$\leq 1.$$

Similarly, we bound the second moment as:

$$\mathbb{E}_t\left[\left(\left(\mathbb{E}_t\left[\widehat{\boldsymbol{\ell}}_t\right] - \widehat{\boldsymbol{\ell}}_t\right)^\top\boldsymbol{x}_t\right)^2\right] = \mathbb{E}_t\left[\left(\left(\boldsymbol{\ell}_t - \widehat{\boldsymbol{\ell}}_t\right)^\top\boldsymbol{x}_t\right)^2\right]$$

$$\leq \mathbb{E}_t\left[\widehat{\boldsymbol{\ell}}_t^\top\boldsymbol{x}_t\right]$$

$$= \boldsymbol{\ell}_t^\top\boldsymbol{x}_t.$$

Thus we can apply the Freedman inequality to attain, with probability at least $1 - \delta$:

$$\sum_{t=1}^{T} (\boldsymbol{\ell}_t - \widehat{\boldsymbol{\ell}}_t)^\top \boldsymbol{x}_t = \mathcal{O}\left(\sqrt{\sum_{t=1}^{T} \boldsymbol{\ell}_t^\top \boldsymbol{x}_t \ln\left(\frac{1}{\delta}\right)} + \ln\left(\frac{1}{\delta}\right)\right).$$

**Bound on the third term** To bound the third term, we again notice that the quantity of interest is a Martingale difference sequence, but we apply a modified version of the Freedman inequality (see [Lee et al., 2020a]).

First we notice that:

$$(\widehat{\boldsymbol{\ell}}_t - \boldsymbol{\ell}_t)^\top \boldsymbol{u} \le \boldsymbol{h}_t^\top \boldsymbol{u} \in [1, T].$$

We now focus on bounding the second moment as follows:

$$\mathbb{E}_t\left[\left((\widehat{\boldsymbol{\ell}}_t - \boldsymbol{\ell}_t)^\top \boldsymbol{u}\right)^2\right] \le \mathbb{E}_t\left[\left(\widehat{\boldsymbol{\ell}}_t^\top \boldsymbol{u}\right)^2\right]$$

$$= \mathbb{E}_t\left[\frac{\ell_t^2(a_t) u^2(a_t)}{x_t^2(a_t)}\right]$$

$$\le \sum_{a=1}^{K} u^2(a) \ell_t(a) h_{T,a}$$

$$\le \boldsymbol{h}_T^\top \boldsymbol{u} \cdot \boldsymbol{\ell}_t^\top \boldsymbol{u}.$$

Thus, with probability at least $1 - \delta$, we have by Theorem 2.2 of [Lee et al., 2020a]:

$$\sum_{t=1}^{T} (\widehat{\boldsymbol{\ell}}_t - \boldsymbol{\ell}_t)^\top \boldsymbol{u} = H\left(\sqrt{8 \sum_{t=1}^{T} \boldsymbol{\ell}_t^\top \boldsymbol{u} \cdot \boldsymbol{h}_T^\top \boldsymbol{u} \ln\left(\frac{H}{\delta}\right)} + 2\boldsymbol{h}_T^\top \boldsymbol{u} \ln\left(\frac{H}{\delta}\right)\right),$$

where $H = \ln\left(\frac{\lceil \log(T) \rceil \lceil 3 \log(T) \rceil}{\delta}\right)$.

**Final result** Combining the previous results and applying a Union Bound, we have, with probability at least $1 - 3\delta$:

$$R_T \le \mathcal{O}\left(\frac{K \ln T}{\eta} + \eta \sum_{T=1}^{T} \ell_t(a_t)\right) - \frac{\boldsymbol{h}_T^\top \boldsymbol{u}}{10\eta \ln T} + \mathcal{O}\left(\sqrt{\sum_{t=1}^{T} \boldsymbol{\ell}_t^\top \boldsymbol{x}_t \ln\left(\frac{1}{\delta}\right)} + \ln\left(\frac{1}{\delta}\right)\right) +$$

$$H\left(\sqrt{8 \sum_{t=1}^{T} \boldsymbol{\ell}_t^\top \boldsymbol{u} \cdot \boldsymbol{h}_T^\top \boldsymbol{u} \ln\left(\frac{H}{\delta}\right)} + 2\boldsymbol{h}_T^\top \boldsymbol{u} \ln\left(\frac{H}{\delta}\right)\right).$$

Now we notice that, applying Freedman inequality, it is easy to show the following bound:

$$\sum_{T=1}^{T} \ell_t(a_t) - \sum_{t=1}^{T} \boldsymbol{\ell}_t^\top \boldsymbol{x}_t \le 2\sqrt{\sum_{t=1}^{T} \boldsymbol{\ell}_t^\top \boldsymbol{x}_t \ln\left(\frac{1}{\delta}\right)} + \ln\left(\frac{1}{\delta}\right),$$

which holds with probability at least $1 - \delta$ and implies, by AM-GM inequality:

$$\sum_{t=1}^{T} \ell_t(a_t) \le 2 \sum_{t=1}^{T} \boldsymbol{\ell}_t^\top \boldsymbol{x}_t + 2 \ln\left(\frac{1}{\delta}\right).$$

Now, going back to the regret bound, it holds, with probability at least $1 - 4\delta$, by Union Bound:

$$R_T \le \mathcal{O}\left(\frac{K \ln T}{\eta} + \eta \sum_{t=1}^{T} \boldsymbol{\ell}_t^\top \boldsymbol{x}_t + \eta \ln\left(\frac{1}{\delta}\right)\right) - \frac{\boldsymbol{h}_T^\top \boldsymbol{u}}{10\eta \ln T} + \mathcal{O}\left(\sqrt{\sum_{t=1}^{T} \boldsymbol{\ell}_t^\top \boldsymbol{x}_t \ln\left(\frac{1}{\delta}\right)} + \ln\left(\frac{1}{\delta}\right)\right)$$

$$+ H\left(\sqrt{8\sum_{t=1}^{T}\boldsymbol{\ell}_t^{\top}\boldsymbol{u}\cdot\boldsymbol{h}_T^{\top}\boldsymbol{u}\ln\left(\frac{H}{\delta}\right)}+2\boldsymbol{h}_T^{\top}\boldsymbol{u}\ln\left(\frac{H}{\delta}\right)\right)$$

$$= \mathcal{O}\left(\frac{K\ln T}{\eta}+\eta\sum_{t=1}^{T}\boldsymbol{\ell}_t^{\top}\boldsymbol{x}_t+\ln\left(\frac{1}{\delta}\right)+\sqrt{\sum_{t=1}^{T}\boldsymbol{\ell}_t^{\top}\boldsymbol{x}_t\ln\left(\frac{1}{\delta}\right)}\right)-\frac{\boldsymbol{h}_T^{\top}\boldsymbol{u}}{10\eta\ln T}$$

$$+ H\left(\sqrt{8\frac{20H\eta\ln T}{20H\eta\ln T}\sum_{t=1}^{T}\boldsymbol{\ell}_t^{\top}\boldsymbol{u}\cdot\boldsymbol{h}_T^{\top}\boldsymbol{u}\ln\left(\frac{H}{\delta}\right)}+2\boldsymbol{h}_T^{\top}\boldsymbol{u}\ln\left(\frac{H}{\delta}\right)\right)$$

$$\leq \mathcal{O}\left(\frac{K\ln T}{\eta}+\eta\sum_{t=1}^{T}\boldsymbol{\ell}_t^{\top}\boldsymbol{x}_t+\ln\left(\frac{1}{\delta}\right)+\sqrt{\sum_{t=1}^{T}\boldsymbol{\ell}_t^{\top}\boldsymbol{x}_t\ln\left(\frac{1}{\delta}\right)}\right)-\frac{\boldsymbol{h}_T^{\top}\boldsymbol{u}}{10\eta\ln T}$$

$$+ 160H^2\eta\ln T\sum_{t=1}^{T}\boldsymbol{\ell}_t^{\top}\boldsymbol{u}\ln\left(\frac{H}{\delta}\right)+\frac{H}{20H\eta\ln T}\cdot\boldsymbol{h}_T^{\top}\boldsymbol{u}+2H\boldsymbol{h}_T^{\top}\boldsymbol{u}\ln\left(\frac{H}{\delta}\right) \tag{9a}$$

$$\leq \mathcal{O}\left(\frac{K\ln T}{\eta}+\eta\sum_{t=1}^{T}\boldsymbol{\ell}_t^{\top}\boldsymbol{x}_t+\ln\left(\frac{1}{\delta}\right)+\sqrt{\sum_{t=1}^{T}\boldsymbol{\ell}_t^{\top}\boldsymbol{x}_t\ln\left(\frac{1}{\delta}\right)}\right)$$

$$+ 160H^2\eta\ln T\sum_{t=1}^{T}\boldsymbol{\ell}_t^{\top}\boldsymbol{u}\ln\left(\frac{H}{\delta}\right) \tag{9b}$$

$$\leq \widetilde{\mathcal{O}}\left(\frac{K}{\eta}+\eta\sum_{t=1}^{T}\boldsymbol{\ell}_t^{\top}\boldsymbol{x}_t+\sqrt{\sum_{t=1}^{T}\boldsymbol{\ell}_t^{\top}\boldsymbol{x}_t\ln\left(\frac{1}{\delta}\right)}+\eta\sum_{t=1}^{T}\boldsymbol{\ell}_t^{\top}\boldsymbol{u}\ln\left(\frac{1}{\delta}\right)\right)$$

$$\leq \widetilde{\mathcal{O}}\left(\frac{K}{\eta}+\eta\sum_{t=1}^{T}\boldsymbol{\ell}_t^{\top}\boldsymbol{x}_t+\sqrt{\sum_{t=1}^{T}\boldsymbol{\ell}_t^{\top}\boldsymbol{x}_t\ln\left(\frac{1}{\delta}\right)}+\eta\sum_{t=1}^{T}\boldsymbol{\ell}_t^{\top}\boldsymbol{x}^*\ln\left(\frac{1}{\delta}\right)\right), \tag{9c}$$

where Inequality (9a) holds by AM-GM inequality, Inequality (9b) holds for $\eta\leq\frac{1}{40H\ln T\ln\left(\frac{H}{\delta}\right)}$ and Inequality (9c) holds after noticing that by definition of $\boldsymbol{u}$, $\sum_{t=1}^{T}\boldsymbol{\ell}_t^{\top}\boldsymbol{u}\leq\sum_{t=1}^{T}\boldsymbol{\ell}_t^{\top}\boldsymbol{x}^*+K$. Since $\eta\leq\frac{1}{2}$, we have:

$$\eta\sum_{t=1}^{T}\boldsymbol{\ell}_t^{\top}\boldsymbol{x}_t\leq\frac{1}{2}R_T+\eta\sum_{t=1}^{T}\boldsymbol{\ell}_t^{\top}\boldsymbol{x}^*,$$

and the regret can be rewritten as:

$$R_T\leq\widetilde{\mathcal{O}}\left(\frac{2K}{\eta}+2\sqrt{\left(\frac{1}{2}R_T+\eta\sum_{t=1}^{T}\boldsymbol{\ell}_t^{\top}\boldsymbol{x}^*\right)\ln\left(\frac{1}{\delta}\right)}+4\eta\sum_{t=1}^{T}\boldsymbol{\ell}_t^{\top}\boldsymbol{x}^*\right).$$

We then set $\eta=\min\left\{\frac{1}{40H\ln T\ln\left(\frac{H}{\delta}\right)},\sqrt{\frac{K}{\sum_{t=1}^{T}\boldsymbol{\ell}_t^{\top}\boldsymbol{x}^*\ln\left(\frac{1}{\delta}\right)}}\right\}$ and we solve the quadratic inequality in $R_T$, obtaining the following regret bound:

$$R_T\leq\widetilde{\mathcal{O}}\left(\sqrt{K\sum_{t=1}^{T}\boldsymbol{\ell}_t^{\top}\boldsymbol{x}^*\ln\left(\frac{1}{\delta}\right)}\right).$$

This concludes the proof. $\qquad\square$

### C.3 Cumulative Violations of COLB

**Theorem 1.** *Let $\delta\in(0,1)$. Then, with probability at least $1-2\delta$, the* COLB *algorithm suffers cumulative positive constraint violations $V_T\leq\mathcal{O}\left(\sqrt{KT\ln\left(TKm/\delta\right)}\right)$.*

*Proof.* First, we underline that the following analysis holds for every constraint $i \in [m]$, including the one being violated the most, *i.e.*, $\widetilde{i} \in \arg\max_{i \in [m]} \sum_{t=1}^{T} \left[ g_i^\top x_t - \alpha_i \right]^+$.

By Lemma 2 we have that, under the clean event, $x_t \in \widetilde{\mathcal{S}}_t$. By construction, this implies $(\widehat{g}_{t-1,i} - \beta_{t-1})^\top x_t \le \alpha_i + \frac{K}{T}$ for every $t \in [T]$. Employing Lemma 1, we get, under the clean event:

$$\left[ g_i^\top x_t - \alpha_i \right]^+ \le \frac{K}{T} + 2\beta_{t-1}^\top x_t.$$

To bound the second term we proceed as follows:

$$\sum_{t=1}^{T} \beta_{t-1}^\top x_t = \sum_{t=1}^{T} \sum_{a=1}^{K} \beta_{t-1}(a) x_t(a)$$

$$\le \sum_{t=1}^{T} \sum_{a=1}^{K} \beta_{t-1}(a) \mathbb{1}_t(a) + \sqrt{2T \log(\delta^{-1})} \tag{10}$$

$$= \sqrt{4 \ln\left(\frac{TKm}{\delta}\right)} \sum_{t=1}^{T} \sum_{a=1}^{K} \frac{\mathbb{1}_t(a)}{\sqrt{\max\{1, N_{t-1}(a)\}}} + \sqrt{2T \ln(\delta^{-1})}$$

$$\le 3\sqrt{4 \ln\left(\frac{TKm}{\delta}\right)} \sum_{a=1}^{K} \sqrt{N_T(a)} + \sqrt{2T \ln(\delta^{-1})} \tag{11}$$

$$\le 3\sqrt{4KT \ln\left(\frac{TKm}{\delta}\right)} + \sqrt{2T \ln(\delta^{-1})}, \tag{12}$$

where Inequality (10) follows from Azuma Inequality, with probability at least $1 - \delta$, Inequality (11) holds since $\sum_{t=1}^{T} \frac{1}{\sqrt{t}} \le 3\sqrt{T}$, and Inequality (12) from Cauchy-Schwarz Inequality and the fact that $\sum_{a=1}^{K} N_T(a) = T$.

Finally, it holds:

$$V_T = \sum_{t=1}^{T} \left[ g_i^\top x_t - \alpha_i \right]^+$$

$$\le \sum_{t=1}^{T} \left( \frac{K}{T} + 2\beta_{t-1}^\top x_t \right)$$

$$\le K + 3\sqrt{4KT \ln\left(\frac{TKm}{\delta}\right)} + \sqrt{2T \ln(\delta^{-1})}.$$

Employing a Union Bound concludes the proof. $\qquad\square$

## D  Omitted Proofs for Hard Constraints

### D.1  Cumulative Regret of SOLB

**Lemma 3.** *Let $\delta \in (0, 1)$ and $\rho \ge \frac{12K}{T}$. Then, with probability at least $1 - 2\delta$, SOLB satisfies:*

$$R_T^\diamond(\ell_{1:T}) \le \mathcal{O}\left( \frac{K}{\rho} \sqrt{\sum_{t=1}^{T} \left( \ell_t^\top (x^\diamond - x^*) \right)^2 \cdot \ln\left(\frac{KTm}{\delta}\right)} + \frac{K}{\rho^6} \ln\left(\frac{KTm}{\delta}\right) \right), \tag{2}$$

*where $R_T^\diamond(\ell_{1:T}) := \sum_{t=1}^{T} \gamma_{t-1} \ell_t^\top (x^\diamond - x^*)$.*

*Proof.* We first split the round in two sets $T_1, T_2$. $T_1$ encompasses the rounds $t \in [T]$ s.t. $\gamma_{t-1} \le 1/2$, $T_2$ the remaining rounds.

**Bound in $T_1$**  We apply the Cauchy–Schwarz inequality obtaining the following bound:

$$\sum_{t \in T_1} \gamma_{t-1} \boldsymbol{\ell}_t^\top (\boldsymbol{x}^\diamond - \boldsymbol{x}^*) \leq \sqrt{\left( \sum_{t \in T_1} \gamma_{t-1}^2 \right) \left( \sum_{t \in T_1} \left( \boldsymbol{\ell}_t^\top (\boldsymbol{x}^\diamond - \boldsymbol{x}^*) \right)^2 \right)}$$

$$= \sqrt{\sum_{t \in T_1} \gamma_{t-1}^2} \cdot \sqrt{\sum_{t=1}^{T} \left( \boldsymbol{\ell}_t^\top (\boldsymbol{x}^\diamond - \boldsymbol{x}^*) \right)^2}.$$

We will now focus on the bounding the sequence $\sum_{t \in T_1} \gamma_{t-1}^2$.

We proceed as follows:

$$\sum_{t \in T_1} \gamma_{t-1}^2 = \sum_{t \in T_1} \max_{i \in \mathcal{M}} \left\{ \frac{\min\{(\widehat{\boldsymbol{g}}_{t-1,i} + \boldsymbol{\beta}_{t-1})^\top \widetilde{\boldsymbol{x}}_t, 1\} - \alpha_i}{\min\{(\widehat{\boldsymbol{g}}_{t-1,i} + \boldsymbol{\beta}_{t-1})^\top \widetilde{\boldsymbol{x}}_t, 1\} - \theta_i} \right\}^2$$

$$\leq \sum_{t \in T_1} \max_{i \in \mathcal{M}} \left\{ \frac{(\widehat{\boldsymbol{g}}_{t-1,i} + \boldsymbol{\beta}_{t-1})^\top \widetilde{\boldsymbol{x}}_t - \alpha_i}{(\widehat{\boldsymbol{g}}_{t-1,i} + \boldsymbol{\beta}_{t-1})^\top \widetilde{\boldsymbol{x}}_t - \theta_i} \right\}^2$$

$$\leq \sum_{t \in T_1} \left( \frac{2 \boldsymbol{\beta}_{t-1}^\top \widetilde{\boldsymbol{x}}_t + \frac{K}{T}}{\rho} \right)^2$$

$$\leq \sum_{t \in T_1} 2 \left( \frac{2 \boldsymbol{\beta}_{t-1}^\top \widetilde{\boldsymbol{x}}_t}{\rho} \right)^2 + \frac{2K^2}{\rho^2},$$

where the second inequality holds by definition of $\rho$ and by Lemma 1, with probability at least $1 - \delta$.

Thus we bound the following quantity:

$$\sum_{t \in T_1} \left( \boldsymbol{\beta}_{t-1}^\top \widetilde{\boldsymbol{x}}_t \right)^2 \leq \sum_{t \in T_1} \left( 2(1 - \gamma_{t-1}) \boldsymbol{\beta}_{t-1}^\top \widetilde{\boldsymbol{x}}_t \right)^2$$

$$\leq 4 \sum_{t=1}^{T} \left( \boldsymbol{\beta}_{t-1}^\top \boldsymbol{x}_t \right)^2$$

$$= 4 \sum_{t=1}^{T} \left( \sum_{a=1}^{K} \sqrt{\frac{4 \ln(TKm/\delta)}{\max\{1, N_{t-1}(a)\}}} x_t(a) \right)^2$$

$$\leq 4K \sum_{t=1}^{T} \sum_{a=1}^{K} \left( \sqrt{\frac{4 \ln(TKm/\delta)}{\max\{1, N_{t-1}(a)\}}} x_t(a) \right)^2$$

$$\leq K 16 \ln(TKm/\delta) \sum_{t=1}^{T} \sum_{a=1}^{K} \frac{1}{\max\{1, N_{t-1}(a)\}} x_t(a)$$

$$= K 16 \ln(TKm/\delta) \sum_{t=1}^{T} \sum_{a=1}^{K} \left( \frac{x_t(a) - \mathbb{1}_t(a)}{\max\{1, N_{t-1}(a)\}} + \frac{\mathbb{1}_t(a)}{\max\{1, N_{t-1}(a)\}} \right),$$

where the first step holds since $\gamma_{t-1} \leq 1/2$.

We can bound the second term as:

$$\sum_{t=1}^{T} \sum_{a=1}^{K} \frac{\mathbb{1}_t(a)}{\max\{1, N_{t-1}(a)\}} \leq K \left( 1 + \sum_{t=1}^{T} \frac{1}{t} \right) \leq 3K + 2K \ln(T).$$

To bound the first term we notice that it is a martingale difference sequence in which any martingale difference is bounded by 1. Thus, we proceed as follows:

$$\mathbb{E}_t \left[ \left( \sum_{a=1}^{K} \frac{x_t(a) - \mathbb{1}_t(a)}{\max\{1, N_{t-1}(a)\}} \right)^2 \right] \leq \mathbb{E}_t \left[ K \sum_{a=1}^{K} \left( \frac{x_t(a) - \mathbb{1}_t(a)}{\max\{1, N_{t-1}(a)\}} \right)^2 \right]$$

$$= K \sum_{a=1}^{K} \frac{\mathbb{E}_t \left[ (x_t(a) - \mathbb{1}_t(a))^2 \right]}{\max\{1, N_{t-1}^2(a)\}}$$

$$= K \sum_{a=1}^{K} \frac{x_t(a)(1 - x_t(a))}{\max\{1, N_{t-1}^2(a)\}}$$

$$\leq K \sum_{a=1}^{K} \frac{x_t(a)}{\max\{1, N_{t-1}(a)\}},$$

and we apply Lemma 9 of [Jin et al., 2020] with $\lambda = 1/2K$ to obtain, with probability at least $1 - \delta$:

$$\sum_{t=1}^{T} \sum_{a=1}^{K} \frac{x_t(a) - \mathbb{1}_t(a)}{\max\{1, N_{t-1}(a)\}} \leq \frac{1}{2} \sum_{t=1}^{T} \sum_{a=1}^{K} \frac{x_t(a)}{\max\{1, N_{t-1}(a)\}} + 2K \ln(1/\delta).$$

Thus, employing a Union Bound, we obtain, with probability at least $1 - 2\delta$:

$$\sum_{t \in T_1} \left( \boldsymbol{\beta}_{t-1}^{\top} \widetilde{\boldsymbol{x}}_t \right)^2 \leq 96 K^2 \ln(TKm/\delta) + 128 K^2 \ln^2(TKm/\delta),$$

and similarly:

$$\sum_{t \in T_1} \gamma_{t-1}^2 \leq \frac{768}{\rho^2} K^2 \ln(TKm/\delta) + \frac{1024}{\rho^2} K^2 \ln^2(TKm/\delta) + \frac{2K^2}{\rho^2}.$$

To conclude, we have, with probability at least $1 - 2\delta$:

$$\sum_{t \in T_1} \gamma_{t-1} \boldsymbol{\ell}_t^{\top} (\boldsymbol{x}^{\diamond} - \boldsymbol{x}^*)$$

$$\leq \sqrt{\sum_{t \in T_1} \gamma_{t-1}^2} \cdot \sqrt{\sum_{t=1}^{T} \left( \boldsymbol{\ell}_t^{\top} (\boldsymbol{x}^{\diamond} - \boldsymbol{x}^*) \right)^2}$$

$$\leq \sqrt{\frac{768}{\rho^2} K^2 \ln(TKm/\delta) + \frac{1024}{\rho^2} K^2 \ln^2(TKm/\delta) + \frac{2K^2}{\rho^2}} \cdot \sqrt{\sum_{t=1}^{T} \left( \boldsymbol{\ell}_t^{\top} (\boldsymbol{x}^{\diamond} - \boldsymbol{x}^*) \right)^2}$$

$$\leq \frac{43 K \ln(TKm/\delta)}{\rho} \sqrt{\sum_{t=1}^{T} \left( \boldsymbol{\ell}_t^{\top} (\boldsymbol{x}^{\diamond} - \boldsymbol{x}^*) \right)^2}.$$

**Bound in $T_2$** We first apply the Hölder inequality to obtain the following bound:

$$\sum_{t \in T_2} \gamma_{t-1} \boldsymbol{\ell}_t^{\top} (\boldsymbol{x}^{\diamond} - \boldsymbol{x}^*) \leq \sum_{t \in T_2} \gamma_{t-1}.$$

To bound the aforementioned terms, we upper bound the cardinality of the set $T_2$. This is done by first bounding the cardinality of the following set:

$$T_3 = \left\{ t \in [T] : \sum_{a=1}^{K} \beta_{t-1}(a) \mathbb{1}_t(a) \geq \frac{\rho^2}{8} \right\}.$$

From the definition we can state the following lower bound:

$$\sum_{t \in T_3} \sum_{a=1}^{K} \beta_{t-1}(a) \mathbb{1}_t(a) \geq |T_3| \frac{\rho^2}{8}.$$

We first bound the quantity $\sum_{t \in T_3} \sum_{a=1}^{K} \beta_{t-1}(a) \mathbb{1}_t(a)$ similarly to what done in Theorem 1 as:

$$\sum_{t \in T_3} \sum_{a=1}^{K} \beta_{t-1}(a) \mathbb{1}_t(a) \leq 3 \sqrt{4K |T_3| \ln \left( \frac{TKm}{\delta} \right)},$$

which holds with probability at least $1 - \delta$. Combining the previous bounds, we obtain:

$$|T_3|\frac{\rho^2}{8} \leq 3\sqrt{4K|T_3|\ln\left(\frac{TKm}{\delta}\right)},$$

which implies:

$$|T_3| \leq \frac{2304}{\rho^4}K\ln\left(\frac{TKm}{\delta}\right).$$

Thus, we employ the reverse Markov inequality to bound the probability that $t \in T_2 \cap T_3$. First we lower bound the following quantity:

$$\mathbb{E}_t\left[\sum_{a=1}^{K}\beta_{t-1}(a)\mathbb{1}_t(a)\right] = \boldsymbol{\beta}_{t-1}^{\top}\boldsymbol{x}_t$$

$$\geq (1 - \gamma_{t-1})\boldsymbol{\beta}_{t-1}^{\top}\widetilde{\boldsymbol{x}}_t$$

$$\geq \frac{\rho}{1+\rho}\boldsymbol{\beta}_{t-1}^{\top}\widetilde{\boldsymbol{x}}_t$$

$$\geq \frac{\rho^2}{4} - \frac{\rho K}{T}$$

$$\geq \frac{\rho^2}{6},$$

where the last steps hold since:

$$\gamma_{t-1} \leq \max_{i\in[m]}\left\{\frac{1-\alpha_i}{1-\theta_i}\right\} = \max_{i\in[m]}\left\{\frac{1-\alpha_i}{1+\rho-\alpha_i}\right\} \leq \frac{1}{1+\rho},$$

$\boldsymbol{\beta}_{t-1}^{\top}\widetilde{\boldsymbol{x}}_t \geq \frac{\rho}{2} - \frac{K}{T}$ when $t \in T_2$, under the clean event and for $\rho \geq \frac{12K}{T}$. We can now employ the reverse Markov inequality to state:

$$\mathbb{P}\left\{\sum_{a=1}^{K}\beta_{t-1}(a)\mathbb{1}_t(a) \geq \frac{\rho^2}{8}\Big|\mathcal{F}_{t-1}\right\} \geq \frac{\frac{\rho^2}{6} - \frac{\rho^2}{8}}{1 - \frac{\rho^2}{8}} \geq \frac{\rho^2}{24}.$$

Employing the equation above we can state that:

$$\frac{2304}{\rho^4}K\ln\left(\frac{TKm}{\delta}\right) \geq |T_3| \geq \frac{\rho^2}{24}|T_2|,$$

from which:

$$|T_2| \leq \frac{55296}{\rho^6}K\ln\left(\frac{TKm}{\delta}\right).$$

To conclude, we have, with probability at least $1 - \delta$, the following bound:

$$\sum_{t\in T_2}\gamma_{t-1}\boldsymbol{\ell}_t^{\top}(\boldsymbol{x}^{\diamond} - \boldsymbol{x}^*) \leq \sum_{t\in T_2}\gamma_{t-1} \leq |T_2| \leq \frac{55296}{\rho^6}K\ln\left(\frac{TKm}{\delta}\right).$$

**Combining everything** Considering the quantity of interest, we have the following bound with probability at least $1 - 2\delta$ by Union Bound:

$$\sum_{t=1}^{T}\gamma_{t-1}\boldsymbol{\ell}_t^{\top}(\boldsymbol{x}^{\diamond} - \boldsymbol{x}^*) = \sum_{t\in T_1}\gamma_{t-1}\boldsymbol{\ell}_t^{\top}(\boldsymbol{x}^{\diamond} - \boldsymbol{x}^*) + \sum_{t\in T_2}\gamma_{t-1}\boldsymbol{\ell}_t^{\top}(\boldsymbol{x}^{\diamond} - \boldsymbol{x}^*)$$

$$\leq \frac{43K\ln(TKm/\delta)}{\rho}\sqrt{\sum_{t=1}^{T}\left(\boldsymbol{\ell}_t^{\top}(\boldsymbol{x}^{\diamond} - \boldsymbol{x}^*)\right)^2} + \frac{55296}{\rho^6}K\ln\left(\frac{TKm}{\delta}\right).$$

This concludes the proof. $\qquad\square$

**Lemma 4.** *Let $\delta \in (0,1)$ and $\eta \leq \frac{\rho}{40H \ln T \ln(H/\delta)}$, where $H := \ln\left(\lceil \ln(T) \rceil \lceil 3\ln(T) \rceil / \delta\right)$. Then, with probability at least $1 - 2\delta$,* SOLB *satisfies:*

$$\widetilde{R}_T(\boldsymbol{\ell}_{1:T}) \leq \mathcal{O}\left(\frac{K}{\eta} + \frac{\eta}{\rho}\sum_{t=1}^{T}\boldsymbol{\ell}_t^{\top}\boldsymbol{x}_t + \frac{K\eta}{\rho}\ln\left(\frac{1}{\delta}\right) + \sqrt{\sum_{t=1}^{T}\boldsymbol{\ell}_t^{\top}\boldsymbol{x}_t \ln\left(\frac{1}{\delta}\right)} + \frac{\eta}{\rho}\sum_{t=1}^{T}\boldsymbol{\ell}_t^{\top}\boldsymbol{x}^* \ln\left(\frac{1}{\delta}\right)\right),$$

*where $\widetilde{R}_T(\boldsymbol{\ell}_{1:T}) = \sum_{t=1}^{T}(1-\gamma_{t-1})\boldsymbol{\ell}_t^{\top}(\widetilde{\boldsymbol{x}}_t - \boldsymbol{x}^*)$.*

*Proof.* Similarly to the analysis employed to prove Theorem 2, we decompose the quantity of interest as follows:

$$\sum_{t=1}^{T}(1-\gamma_{t-1})\boldsymbol{\ell}_t^{\top}(\widetilde{\boldsymbol{x}}_t - \boldsymbol{x}^*)$$

$$\leq \sum_{t=1}^{T}(1-\gamma_{t-1})\widehat{\boldsymbol{\ell}}_t^{\top}(\widetilde{\boldsymbol{x}}_t - \boldsymbol{u}) + \sum_{t=1}^{T}(1-\gamma_{t-1})(\boldsymbol{\ell}_t - \widehat{\boldsymbol{\ell}}_t)^{\top}\widetilde{\boldsymbol{x}}_t + \sum_{t=1}^{T}(1-\gamma_{t-1})(\widehat{\boldsymbol{\ell}}_t - \boldsymbol{\ell}_t)^{\top}\boldsymbol{u}$$

$$+ \sum_{t=1}^{T}\boldsymbol{\ell}_t^{\top}(\boldsymbol{u} - \boldsymbol{x}^*)$$

$$\leq \sum_{t=1}^{T}(1-\gamma_{t-1})\widehat{\boldsymbol{\ell}}_t^{\top}(\widetilde{\boldsymbol{x}}_t - \boldsymbol{u}) + \sum_{t=1}^{T}(1-\gamma_{t-1})(\boldsymbol{\ell}_t - \widehat{\boldsymbol{\ell}}_t)^{\top}\widetilde{\boldsymbol{x}}_t + \sum_{t=1}^{T}(1-\gamma_{t-1})(\widehat{\boldsymbol{\ell}}_t - \boldsymbol{\ell}_t)^{\top}\boldsymbol{u} + K.$$

We proceed bounding each term separately.

**Bound on the first term** To bound the first term, we can apply a similar analysis to the one of Lemma 5, since, $\widetilde{\boldsymbol{x}}_t$ is played independently on $\gamma_{t-1}$ except for the loss estimator, to attain, under the clean event:

$$\sum_{t=1}^{T}(1-\gamma_{t-1})\widehat{\boldsymbol{\ell}}_t^{\top}(\widetilde{\boldsymbol{x}}_t - \boldsymbol{u})$$

$$\leq \mathcal{O}\left(\frac{K\ln T}{\eta}\right) - \min_{t\in[T]}(1-\gamma_{t-1})\frac{\boldsymbol{h}_T^{\top}\boldsymbol{u}}{10\eta\ln T} + \sum_{t=1}^{T}(1-\gamma_{t-1})\sum_{a=1}^{K}\eta_{t,a}\widetilde{x}_t^2(a)\widehat{\ell}_t^{\,2}(a)$$

$$\leq \mathcal{O}\left(\frac{K\ln T}{\eta}\right) - \rho\frac{\boldsymbol{h}_T^{\top}\boldsymbol{u}}{10\eta\ln T} + \sum_{t=1}^{T}(1-\gamma_{t-1})\sum_{a=1}^{K}\eta_{t,a}\widetilde{x}_t^2(a)\widehat{\ell}_t^{\,2}(a).$$

To bound the last term, we proceed as follows:

$$(1-\gamma_{t-1})\eta_{t,a}\widetilde{x}_t^2(a)\widehat{\ell}_t^{\,2}(a) = (1-\gamma_{t-1})\eta_{t,a}\widetilde{x}_t^2(a)\frac{\ell_t^2(a)}{x_t^2(a)}\mathbb{1}\{a_t = a\}$$

$$= \frac{1}{1-\gamma_{t-1}}(1-\gamma_{t-1})^2\eta_{t,a}\widetilde{x}_t^2(a)\frac{\ell_t^2(a)}{x_t^2(a)}\mathbb{1}\{a_t = a\}$$

$$\leq \frac{1}{\rho}\eta_{t,a}x_t^2(a)\frac{\ell_t^2(a)}{x_t^2(a)}\mathbb{1}\{a_t = a\}$$

$$\leq \frac{1}{\rho}\eta_{t,a_t}\ell_t^2(a_t)$$

$$\leq \frac{1}{\rho}\eta_{t,a_t}\ell_t(a_t)$$

$$\leq \frac{1}{\rho}\eta_{T,a_t}\ell_t(a_t)$$

$$\leq \frac{5}{\rho}\eta\ell_t(a_t).$$

Thus, we obtain the following final bound, which holds with probability at least $1 - \delta$:

$$\sum_{t=1}^{T}(1-\gamma_{t-1})\widehat{\boldsymbol{\ell}}_t^{\top}(\widetilde{\boldsymbol{x}}_t - \boldsymbol{u}) \leq \mathcal{O}\left(\frac{K\ln T}{\eta} + \frac{\eta}{\rho}\sum_{t=1}^{T}\ell_t(a_t)\right) - \rho\frac{\boldsymbol{h}_T^{\top}\boldsymbol{u}}{10\eta\ln T}.$$

**Bound on the second term**   To bound the second term we first notice that $(1 - \gamma_{t-1})\widetilde{\boldsymbol{x}}_t \leq \boldsymbol{x}_t$. Thus, we proceed similarly to Theorem 2, noticing that the quantity of interest is Martingale difference sequence, where any difference is bounded as:

$$
\left| (1 - \gamma_{t-1}) \left( \mathbb{E}_t \left[ \widehat{\boldsymbol{\ell}}_t \right] - \widehat{\boldsymbol{\ell}}_t \right)^\top \widetilde{\boldsymbol{x}}_t \right| \leq (1 - \gamma_{t-1}) \widehat{\boldsymbol{\ell}}_t^\top \widetilde{\boldsymbol{x}}_t
$$

$$
\leq \widehat{\boldsymbol{\ell}}_t^\top \boldsymbol{x}_t
$$

$$
= \sum_{a=1}^K x_t(a) \frac{\ell_t(a)}{x_t(a)} \mathbb{1}\{a_t = a\}
$$

$$
\leq 1.
$$

Furthermore, we bound the second moment as:

$$
\mathbb{E}_t \left[ \left( (1 - \gamma_{t-1}) \left( \mathbb{E}_t \left[ \widehat{\boldsymbol{\ell}}_t \right] - \widehat{\boldsymbol{\ell}}_t \right)^\top \widetilde{\boldsymbol{x}}_t \right)^2 \right] = \mathbb{E}_t \left[ \left( (1 - \gamma_{t-1}) \left( \boldsymbol{\ell}_t - \widehat{\boldsymbol{\ell}}_t \right)^\top \widetilde{\boldsymbol{x}}_t \right)^2 \right]
$$

$$
\leq \mathbb{E}_t \left[ \left( (1 - \gamma_{t-1}) \widehat{\boldsymbol{\ell}}_t^\top \widetilde{\boldsymbol{x}}_t \right)^2 \right]
$$

$$
\leq \mathbb{E}_t \left[ \left( \widehat{\boldsymbol{\ell}}_t^\top \boldsymbol{x}_t \right)^2 \right]
$$

$$
\leq \mathbb{E}_t \left[ \widehat{\boldsymbol{\ell}}_t^\top \boldsymbol{x}_t \right]
$$

$$
= \boldsymbol{\ell}_t^\top \boldsymbol{x}_t.
$$

Thus we can apply the Freedman inequality to attain, with probability at least $1 - \delta$:

$$
\sum_{t=1}^T (1 - \gamma_{t-1})(\boldsymbol{\ell}_t - \widehat{\boldsymbol{\ell}}_t)^\top \widetilde{\boldsymbol{x}}_t \leq \mathcal{O} \left( \sqrt{\sum_{t=1}^T \boldsymbol{\ell}_t^\top \boldsymbol{x}_t \ln \left( \frac{1}{\delta} \right)} + \ln \left( \frac{1}{\delta} \right) \right).
$$

**Bound on the third term**   To bound the third term, we notice that the quantity of interest is a Martingale difference sequence. To apply the modified version of the Freedman inequality (see [Lee et al., 2020a]), we notice that:

$$
(1 - \gamma_{t-1})(\widehat{\boldsymbol{\ell}}_t - \boldsymbol{\ell}_t)^\top \boldsymbol{u} \leq (1 - \gamma_{t-1}) \sum_{a=1}^K \frac{1}{x_t(a)} u(a)
$$

$$
= (1 - \gamma_{t-1}) \sum_{a=1}^K \frac{1}{\gamma_{t-1} x^\diamond(a) + (1 - \gamma_{t-1}) \widetilde{x}_t(a)} u(a)
$$

$$
\leq (1 - \gamma_{t-1}) \sum_{a=1}^K \frac{1}{(1 - \gamma_{t-1}) \widetilde{x}_t(a)} u(a)
$$

$$
\leq \sum_{a=1}^K \frac{1}{\min_{\tau \in [t]} \widetilde{x}_\tau(a)} u(a)
$$

$$
\leq \boldsymbol{h}_t^\top \boldsymbol{u} \in [1, T].
$$

We now focus on bounding the second moment as follows:

$$
\mathbb{E}_t \left[ \left( (1 - \gamma_{t-1})(\widehat{\boldsymbol{\ell}}_t - \boldsymbol{\ell}_t)^\top \boldsymbol{u} \right)^2 \right] \leq \mathbb{E}_t \left[ \left( (1 - \gamma_{t-1}) \widehat{\boldsymbol{\ell}}_t^\top \boldsymbol{u} \right)^2 \right]
$$

$$
= \mathbb{E}_t \left[ (1 - \gamma_{t-1})^2 \frac{\ell_t^2(a_t) u^2(a_t)}{x_t^2(a_t)} \right]
$$

$$
\leq \mathbb{E}_t \left[ (1 - \gamma_{t-1})^2 \frac{\ell_t^2(a_t) u^2(a_t)}{(1 - \gamma_t)^2 \widetilde{x}_t^2(a_t)} \right]
$$

$$\leq \sum_{a=1}^{K} u^2(a)\ell_t(a)h_{T,a}$$

$$\leq \boldsymbol{h}_T^\top \boldsymbol{u} \cdot \boldsymbol{\ell}_t^\top \boldsymbol{u}.$$

Thus, with probability at least $1 - \delta$, we have by Theorem 2.2 of [Lee et al., 2020a]:

$$\sum_{t=1}^{T}(1 - \gamma_{t-1})(\widehat{\boldsymbol{\ell}}_t - \boldsymbol{\ell}_t)^\top \boldsymbol{u} = H\left(\sqrt{8\sum_{t=1}^{T}\boldsymbol{\ell}_t^\top \boldsymbol{u} \cdot \boldsymbol{h}_T^\top \boldsymbol{u} \ln\left(\frac{H}{\delta}\right)} + \boldsymbol{h}_T^\top \boldsymbol{u} \ln\left(\frac{H}{\delta}\right)\right),$$

where $H = \ln\left(\frac{\lceil \ln(T)\rceil \lceil 3\ln(T)\rceil}{\delta}\right)$.

**Final result**  Combining the previous equations, we get, with probability at least $1 - 3\delta$, by Union Bound, the following bound:

$$\sum_{t=1}^{T}(1 - \gamma_{t-1})\boldsymbol{\ell}_t^\top(\widetilde{\boldsymbol{x}}_t - \boldsymbol{x}^*)$$

$$\leq \mathcal{O}\left(\frac{K\ln T}{\eta} + \frac{\eta}{\rho}\sum_{t=1}^{T}\ell_t(a_t)\right) - \rho\frac{\boldsymbol{h}_T^\top \boldsymbol{u}}{10\eta\ln T} + \mathcal{O}\left(\sqrt{\sum_{t=1}^{T}\boldsymbol{\ell}_t^\top \boldsymbol{x}_t \ln\left(\frac{1}{\delta}\right)} + \ln\left(\frac{1}{\delta}\right)\right)$$

$$+ H\left(\sqrt{8\sum_{t=1}^{T}\boldsymbol{\ell}_t^\top \boldsymbol{u} \cdot \boldsymbol{h}_T^\top \boldsymbol{u} \ln\left(\frac{H}{\delta}\right)} + \boldsymbol{h}_T^\top \boldsymbol{u} \ln\left(\frac{H}{\delta}\right)\right) + K$$

$$\leq \mathcal{O}\left(\frac{K\ln T}{\eta} + \frac{\eta}{\rho}\sum_{t=1}^{T}\ell_t(a_t) + \sqrt{\sum_{t=1}^{T}\boldsymbol{\ell}_t^\top \boldsymbol{x}_t \ln\left(\frac{1}{\delta}\right)}\right) - \rho\frac{\boldsymbol{h}_T^\top \boldsymbol{u}}{10\eta\ln T}$$

$$+ H\left(\sqrt{8\sum_{t=1}^{T}\boldsymbol{\ell}_t^\top \boldsymbol{u} \cdot \boldsymbol{h}_T^\top \boldsymbol{u} \ln\left(\frac{H}{\delta}\right)} + \boldsymbol{h}_T^\top \boldsymbol{u} \ln\left(\frac{H}{\delta}\right)\right),$$

where $H = \ln\left(\frac{\lceil \ln(T)\rceil \lceil 3\ln(T)\rceil}{\delta}\right)$.

Finally, we proceed similarly to Theorem 2, obtaining:

$$\sum_{t=1}^{T}(1 - \gamma_{t-1})\boldsymbol{\ell}_t^\top(\widetilde{\boldsymbol{x}}_t - \boldsymbol{x}^*)$$

$$\leq \mathcal{O}\left(\frac{K\ln T}{\eta} + \frac{2\eta}{\rho}\sum_{t=1}^{T}\boldsymbol{\ell}_t^\top \boldsymbol{x}_t + \frac{2\eta}{\rho}\ln\left(\frac{1}{\delta}\right) + \sqrt{\sum_{t=1}^{T}\boldsymbol{\ell}_t^\top \boldsymbol{x}_t \ln\left(\frac{1}{\delta}\right)}\right) - \rho\frac{\boldsymbol{h}_T^\top \boldsymbol{u}}{10\eta\ln T}$$

$$+ H\left(\sqrt{8\sum_{t=1}^{T}\boldsymbol{\ell}_t^\top \boldsymbol{u} \cdot \boldsymbol{h}_T^\top \boldsymbol{u} \ln\left(\frac{H}{\delta}\right)} + \boldsymbol{h}_T^\top \boldsymbol{u} \ln\left(\frac{H}{\delta}\right)\right)$$

$$= \mathcal{O}\left(\frac{K\ln T}{\eta} + \frac{2\eta}{\rho}\sum_{t=1}^{T}\boldsymbol{\ell}_t^\top \boldsymbol{x}_t + \frac{2\eta}{\rho}\ln\left(\frac{1}{\delta}\right) + \sqrt{\sum_{t=1}^{T}\boldsymbol{\ell}_t^\top \boldsymbol{x}_t \ln\left(\frac{1}{\delta}\right)}\right) - \rho\frac{\boldsymbol{h}_T^\top \boldsymbol{u}}{10\eta\ln T}$$

$$+ H\left(\sqrt{8\frac{20\rho H\eta\ln T}{20\rho H\eta\ln T}\sum_{t=1}^{T}\boldsymbol{\ell}_t^\top \boldsymbol{u} \cdot \boldsymbol{h}_T^\top \boldsymbol{u} \ln\left(\frac{H}{\delta}\right)} + \boldsymbol{h}_T^\top \boldsymbol{u} \ln\left(\frac{H}{\delta}\right)\right)$$

$$\leq \mathcal{O}\left(\frac{K\ln T}{\eta} + \frac{2\eta}{\rho}\sum_{t=1}^{T}\boldsymbol{\ell}_t^\top \boldsymbol{x}_t + \frac{2\eta}{\rho}\ln\left(\frac{1}{\delta}\right) + \sqrt{\sum_{t=1}^{T}\boldsymbol{\ell}_t^\top \boldsymbol{x}_t \ln\left(\frac{1}{\delta}\right)}\right) - \rho\frac{\boldsymbol{h}_T^\top \boldsymbol{u}}{10\eta\ln T}$$

$$+ \frac{160 H^2 \eta \ln T}{\rho} \sum_{t=1}^{T} \boldsymbol{\ell}_t^\top \boldsymbol{u} \ln \left( \frac{H}{\delta} \right) + \frac{\rho H}{20 H \eta \ln T} \boldsymbol{h}_T^\top \boldsymbol{u} + H \boldsymbol{h}_T^\top \boldsymbol{u} \ln \left( \frac{H}{\delta} \right)$$

$$\leq \widetilde{\mathcal{O}} \left( \frac{K}{\eta} + \frac{\eta}{\rho} \sum_{t=1}^{T} \boldsymbol{\ell}_t^\top \boldsymbol{x}_t + \frac{K\eta}{\rho} \ln \left( \frac{1}{\delta} \right) + \sqrt{\sum_{t=1}^{T} \boldsymbol{\ell}_t^\top \boldsymbol{x}_t \ln \left( \frac{1}{\delta} \right)} + \frac{\eta}{\rho} \sum_{t=1}^{T} \boldsymbol{\ell}_t^\top \boldsymbol{x}^* \ln \left( \frac{1}{\delta} \right) \right),$$

where the first step holds by Freedman inequality and a union bound, setting the confidence to $1 - 4\delta$, and AM-GM inequality, the third step by AM-GM inequality and the last step holds for $\eta \leq \frac{\rho}{40 H \ln T \ln\left( \frac{H}{\delta} \right)}$.

This concludes the proof. $\qquad \square$

**Theorem 4.** *Let $\delta \in (0,1)$, $\rho \geq \frac{12K}{T}$, and $\eta = \min \left\{ \frac{\rho}{40 H \ln T \ln(H/\delta)}, \sqrt{K / \sum_{t=1}^{T} \boldsymbol{\ell}_t^\top \boldsymbol{x}^* \ln(1/\delta)} \right\}$, where $H := \ln \left( \lceil \ln(T) \rceil \lceil 3 \ln(T) \rceil / \delta \right)$. Then,* SOLB *suffers a cumulative regret bounded as:*

$$R_T(\boldsymbol{\ell}_{1:T}) \leq \widetilde{\mathcal{O}} \Bigg( \underbrace{\frac{K \ln (1/\delta)}{\rho} \sqrt{\sum_{t=1}^{T} \left( \boldsymbol{\ell}_t^\top (\boldsymbol{x}^\diamond - \boldsymbol{x}^*) \right)^2}}_{\textit{(A) Safety Complexity}} + \underbrace{\frac{1}{\rho} \sqrt{K \sum_{t=1}^{T} \boldsymbol{\ell}_t^\top \boldsymbol{x}^* \ln \left( \frac{1}{\delta} \right)}}_{\textit{(B) Bandit Complexity}} \Bigg), \qquad (3)$$

*where $\widetilde{\mathcal{O}}$ hides universal constants and logarithmic terms not depending on $\delta$.*

*Proof.* We first notice that the regret can be decomposed as:

$$R_T := \sum_{t=1}^{T} \boldsymbol{\ell}_t^\top \boldsymbol{x}_t - \boldsymbol{\ell}_t^\top \boldsymbol{x}^*$$

$$= \sum_{t=1}^{T} \gamma_{t-1} \boldsymbol{\ell}_t^\top (\boldsymbol{x}^\diamond - \boldsymbol{x}^*) + \sum_{t=1}^{T} (1 - \gamma_{t-1}) \boldsymbol{\ell}_t^\top (\widetilde{\boldsymbol{x}}_t - \boldsymbol{x}^*).$$

Employing Lemma 3, Lemma 4 and a Union Bound, we have, with probability at least $1 - 5\delta$:

$$R_T \leq \frac{43 K \ln(TKm/\delta)}{\rho} \sqrt{\sum_{t=1}^{T} \left( \boldsymbol{\ell}_t^\top (\boldsymbol{x}^\diamond - \boldsymbol{x}^*) \right)^2} + \frac{55296}{\rho^6} K \ln \left( \frac{TKm}{\delta} \right)$$

$$+ \widetilde{\mathcal{O}} \left( \frac{K}{\eta} + \frac{\eta}{\rho} \sum_{t=1}^{T} \boldsymbol{\ell}_t^\top \boldsymbol{x}_t + \frac{K\eta}{\rho} \ln \left( \frac{1}{\delta} \right) + \sqrt{\sum_{t=1}^{T} \boldsymbol{\ell}_t^\top \boldsymbol{x}_t \ln \left( \frac{1}{\delta} \right)} + \frac{\eta}{\rho} \sum_{t=1}^{T} \boldsymbol{\ell}_t^\top \boldsymbol{x}^* \ln \left( \frac{1}{\delta} \right) \right)$$

$$= \widetilde{\mathcal{O}} \left( \frac{K}{\rho} \sqrt{\sum_{t=1}^{T} \left( \boldsymbol{\ell}_t^\top (\boldsymbol{x}^\diamond - \boldsymbol{x}^*) \right)^2} + \frac{K}{\eta} + \frac{\eta}{\rho} \sum_{t=1}^{T} \boldsymbol{\ell}_t^\top \boldsymbol{x}_t + \frac{K\eta}{\rho} \ln \left( \frac{1}{\delta} \right) + \sqrt{\sum_{t=1}^{T} \boldsymbol{\ell}_t^\top \boldsymbol{x}_t \ln \left( \frac{1}{\delta} \right)} \right.$$

$$\left. + \frac{\eta}{\rho} \sum_{t=1}^{T} \boldsymbol{\ell}_t^\top \boldsymbol{x}^* \ln \left( \frac{1}{\delta} \right) \right).$$

Since $\eta \leq \frac{\rho}{2}$, we have:

$$\frac{\eta}{\rho} \sum_{t=1}^{T} \boldsymbol{\ell}_t^\top \boldsymbol{x}_t \leq \frac{1}{2} R_T + \frac{\eta}{\rho} \sum_{t=1}^{T} \boldsymbol{\ell}_t^\top \boldsymbol{x}^*,$$

and the regret can be rewritten as:

$$R_T \leq \widetilde{\mathcal{O}} \left( \frac{K}{\rho} \sqrt{\sum_{t=1}^{T} \left( \boldsymbol{\ell}_t^\top (\boldsymbol{x}^\diamond - \boldsymbol{x}^*) \right)^2} + \frac{2K}{\eta} \right.$$

$$+2\sqrt{\left(\frac{1}{2}R_T + \frac{\eta}{\rho}\sum_{t=1}^{T}\boldsymbol{\ell}_t^\top\boldsymbol{x}^*\right)\ln\left(\frac{1}{\delta}\right) + \frac{4\eta}{\rho}\sum_{t=1}^{T}\boldsymbol{\ell}_t^\top\boldsymbol{x}^*\ln\left(\frac{1}{\delta}\right)}.$$

We then set $\eta = \min\left\{\frac{\rho}{40H\ln T\ln\left(\frac{H}{\delta}\right)}, \sqrt{\frac{K}{\sum_{t=1}^{T}\boldsymbol{\ell}_t^\top\boldsymbol{x}^*\ln\left(\frac{1}{\delta}\right)}}\right\}$ and we solve the quadratic inequality in $R_T$, obtaining the following regret bound:

$$R_T \leq \tilde{\mathcal{O}}\left(\frac{K}{\rho}\sqrt{\sum_{t=1}^{T}\left(\boldsymbol{\ell}_t^\top(\boldsymbol{x}^\diamond - \boldsymbol{x}^*)\right)^2} + \frac{1}{\rho}\sqrt{K\sum_{t=1}^{T}\boldsymbol{\ell}_t^\top\boldsymbol{x}^*\ln\left(\frac{1}{\delta}\right)}\right).$$

This concludes the proof. $\qquad\square$

### D.2 Safety Property of SOLB

**Theorem 3.** *Let $\delta \in (0,1)$. With probability at least $1 - \delta$, SOLB guarantees that $\boldsymbol{g}_i^\top\boldsymbol{x}_t \leq \alpha_i$ holds for every constraint $i \in [m]$ and round $t \in [T]$.*

*Proof.* To prove the result, we consider separately the case in which $\gamma_t = 0$ and $\gamma_t \in (0,1)$.

When $\gamma_t = 0$, by construction, it holds that $\forall i \in [m] : (\hat{\boldsymbol{g}}_{t,i} + \boldsymbol{\beta}_t)^\top\tilde{\boldsymbol{x}}_{t+1} \leq \alpha_i$ and $\boldsymbol{x}_{t+1} = \tilde{\boldsymbol{x}}_{t+1}$. Thus, we have:

$$\alpha_i \geq (\hat{\boldsymbol{g}}_{t,i} + \boldsymbol{\beta}_t)^\top\tilde{\boldsymbol{x}}_{t+1}$$
$$= (\hat{\boldsymbol{g}}_{t,i} + \boldsymbol{\beta}_t)^\top\boldsymbol{x}_{t+1}$$
$$\geq \boldsymbol{g}_i^\top\boldsymbol{x}_{t+1},$$

where the last step holds thank to Lemma 1 with probability at least $1 - \delta$.

When $\gamma_t = (0,1)$, $\gamma_t$ can be selected either as $\max_{i\in[m]}\frac{1-\alpha_i}{1-\theta_i}$ or as $\max_{i\in\mathcal{M}}\frac{(\hat{\boldsymbol{g}}_{t,i}+\boldsymbol{\beta}_t)^\top\tilde{\boldsymbol{x}}_{t+1}-\alpha_i}{(\hat{\boldsymbol{g}}_{t,i}+\boldsymbol{\beta}_t)^\top\tilde{\boldsymbol{x}}_{t+1}-\theta_i}$. For simplicity, we study each constraint separately, removing the $\max$ operator. In the first case, it holds:

$$\boldsymbol{g}_i^\top\boldsymbol{x}_{t+1} = \boldsymbol{g}_i^\top(\gamma_t\boldsymbol{x}^\diamond + (1-\gamma_t)\tilde{\boldsymbol{x}}_{t+1})$$
$$\leq \gamma_t\theta_i + (1-\gamma_t)$$
$$= \frac{1-\alpha_i}{1-\theta_i}(\theta_i - 1) + 1$$
$$= \frac{\alpha_i - 1}{\theta_i - 1}(\theta_i - 1) + 1$$
$$= \alpha_i.$$

Similarly, in the latter case, it holds:

$$\boldsymbol{g}_i^\top\boldsymbol{x}_{t+1} = \boldsymbol{g}_i^\top(\gamma_t\boldsymbol{x}^\diamond + (1-\gamma_t)\tilde{\boldsymbol{x}}_{t+1})$$
$$= \gamma_t\theta_i + (1-\gamma_t)\boldsymbol{g}_i^\top\tilde{\boldsymbol{x}}_{t+1}$$
$$\leq \gamma_t\theta_i + (1-\gamma_t)(\hat{\boldsymbol{g}}_{t,i} + \boldsymbol{\beta}_t)^\top\tilde{\boldsymbol{x}}_{t+1}$$
$$= \gamma_t(\theta_i - (\hat{\boldsymbol{g}}_{t,i} + \boldsymbol{\beta}_t)^\top\tilde{\boldsymbol{x}}_{t+1}) + (\hat{\boldsymbol{g}}_{t,i} + \boldsymbol{\beta}_t)^\top\tilde{\boldsymbol{x}}_{t+1}$$
$$= \frac{(\hat{\boldsymbol{g}}_{t,i} + \boldsymbol{\beta}_t)^\top\tilde{\boldsymbol{x}}_{t+1} - \alpha_i}{(\hat{\boldsymbol{g}}_{t,i} + \boldsymbol{\beta}_t)^\top\tilde{\boldsymbol{x}}_{t+1} - \theta_i}(\theta_i - (\hat{\boldsymbol{g}}_{t,i} + \boldsymbol{\beta}_t)^\top\tilde{\boldsymbol{x}}_{t+1}) + (\hat{\boldsymbol{g}}_{t,i} + \boldsymbol{\beta}_t)^\top\tilde{\boldsymbol{x}}_{t+1}$$
$$= \frac{\alpha_i - (\hat{\boldsymbol{g}}_{t,i} + \boldsymbol{\beta}_t)^\top\tilde{\boldsymbol{x}}_{t+1}}{\theta_i - (\hat{\boldsymbol{g}}_{t,i} + \boldsymbol{\beta}_t)^\top\tilde{\boldsymbol{x}}_{t+1}}(\theta_i - (\hat{\boldsymbol{g}}_{t,i} + \boldsymbol{\beta}_t)^\top\tilde{\boldsymbol{x}}_{t+1}) + (\hat{\boldsymbol{g}}_{t,i} + \boldsymbol{\beta}_t)^\top\tilde{\boldsymbol{x}}_{t+1}$$
$$= \alpha_i.$$

Observing that, thanks to the convexity of the estimated feasible set, the $\max$ operator allows the aforementioned reasoning to hold for each constraint $i \in [m]$, concludes the proof. $\qquad\square$

| | $\ell_t(a_1)$ | $\ell_t(a_2)$ | $\ell_t(a_3)$ | $g_t(a_1)$ | $g_t(a_2)$ | $g_t(a_3)$ |
|---|---|---|---|---|---|---|
| $\boldsymbol{\nu}_1$ | $W_t/2$ | $W_t/2$ | $(W_t+\Delta)/2$ | $D_t$ | $D_t$ | $C_t$ |
| $\boldsymbol{\nu}_2$ | $W_t/2$ | $Y_t/2$ | $(W_t+\Delta)/2$ | $B_t$ | $B_t$ | $C_t$ |
| $\boldsymbol{\nu}_3$ | $Y_t/2$ | $W_t/2$ | $(W_t+\Delta)/2$ | $B_t$ | $B_t$ | $C_t$ |
| $\boldsymbol{\nu}_4$ | $Y_t/2$ | $Y_t/2$ | $(W_t+\Delta)/2$ | $B_t$ | $B_t$ | $C_t$ |

Table 1: Summary of the losses (first three columns) and constraints costs (last three columns) associated to each of the four instances.

## D.3 Regret Lower Bound in MABs with Hard Constraints

**Theorem 5.** *Let $K \geq 2$, $T \geq \max\left\{2, (11 + \ln T)\left(\frac{8}{3}\right)^2\right\}$, and $\omega \in \left[\frac{1}{T}\left(\frac{11}{2} + \ln T\right), \frac{1}{2}\right]$. Then for every randomized algorithm, we have $\sup_{\boldsymbol{\ell}_{1:T} \in \mathcal{B}_{\omega,\Delta,T}} \mathbb{E}[R_T(\boldsymbol{\ell}_{1:T})] \geq \Omega\left(\frac{\Delta}{\rho}\sqrt{T} + \sqrt{\omega T}\right)$, where the expectation is taken with respect to the internal randomization of the algorithm.*

*Proof.* We split the proof in two parts: first, we prove a $\Omega\left(\sqrt{\omega T} + \Delta\sqrt{T}/\rho\right)$ lower bound for the expected regret of any randomized algorithm, when the losses are stochastic. In a stochastic setting, $\omega$ represents the double of the expected value of the loss of the best strategy, while $\Delta$ the double of the expected value of the difference between the strictly safe strategy and the benchmark. Second, we show that there exists at least a sequence of loss belonging to $\mathcal{B}_{\omega,\Delta,T}$ such that the lower bound from the first step holds.

**Step 1** Let $B(\omega)$ indicate a Bernoulli probability distribution with mean $\omega \in (0, 1)$. We start by introducing four instances of the hard constrained bandit problem where both losses and constraints costs are stochastic. To do so, we assume that both losses and constraints are sampled in advance, and introduce the following auxiliary sequences, for all $t \in [T]$:

$$W_t \sim B(\omega),$$
$$Y_t \sim B(\omega + \psi),$$
$$B_t \sim B(1/2),$$
$$C_t \sim B(1/2 - \rho),$$
$$D_t \sim B(1/2 + \epsilon),$$

where $\omega \in (0, \frac{1}{2})$, $\psi \in (0, 1 - \omega)$ and $\rho, \epsilon \in (0, \frac{1}{2})$. We consider four instances $\{\boldsymbol{\nu}_i\}_{i=1}^4$, each with $K = 3$ actions, namely $a_1, a_2$ and $a_3$. In Table 1, we summarize how losses and constraints costs are generated for each action and in each instance. Instance $\boldsymbol{\nu}_1$ is the only one having different constraints costs, while action $a_3$ has, for every $t \in [T]$, the same loss in every instance. In all instances, $\boldsymbol{x}^\diamond = (0, 0, 1)$ is the only strictly safe strategy.

We start by considering $\boldsymbol{\nu}_1$: in order to be safe with high probability, for a given confidence level $\delta \in (0, 1)$, any algorithm must satisfy:

$$\mathbb{P}_{\boldsymbol{\nu}_1}\left(\forall t \in [T] : x_t(a_3) \geq \frac{\epsilon}{\epsilon + \rho}\right) \geq 1 - \delta,$$

where $\boldsymbol{x}_t$ is the strategy of the algorithm at time $t$, and $\mathbb{P}_{\boldsymbol{\nu}_1}$ is the probability measure of instance $\boldsymbol{\nu}_1$ which encompasses the randomness of both environment and algorithm. As a consequence, we have:

$$\mathbb{P}_{\boldsymbol{\nu}_1}\left(\sum_{t=1}^{T} x_t(a_3) \geq T\frac{\epsilon}{\epsilon + \rho}\right) \geq 1 - \delta.$$

We now leverage Pinsker's inequality to relate the probability measures $\mathbb{P}_{\boldsymbol{\nu}_1}$ and $\mathbb{P}_{\boldsymbol{\nu}_j}$, with $j \in \{2, 3, 4\}$, as follows:

$$\mathbb{P}_{\boldsymbol{\nu}_j}\left(\sum_{t=1}^{T} x_t(a_3) \geq T\frac{\epsilon}{\epsilon + \rho}\right) \geq \mathbb{P}_{\boldsymbol{\nu}_1}\left(\sum_{t=1}^{T} x_t(a_3) \geq T\frac{\epsilon}{\epsilon + \rho}\right) - \sqrt{\frac{1}{2}\mathrm{KL}_T(\boldsymbol{\nu}_j, \boldsymbol{\nu}_1)},$$

where $\text{KL}_T(\boldsymbol{\nu}_j, \boldsymbol{\nu}_1)$ is the KL-divergence between the probability measures $\mathbb{P}_{\boldsymbol{\nu}_j}$ and $\mathbb{P}_{\boldsymbol{\nu}_1}$ after $T$ rounds of history.

Using the KL decomposition argument from Lemma 1 of [Gerchinovitz and Lattimore, 2016] (which holds for correlated losses, as in our case), and by upper bounding the KL between two Bernoulli r.v.s using the $\chi^2$-divergence (see Lemma 2.8 from Tsybakov [2008]), for every $j \in \{2, 3, 4\}$, we have

$$\text{KL}_T(\boldsymbol{\nu}_j, \boldsymbol{\nu}_1) \leq T(2\text{KL}(B(\omega + \psi), B(\omega)) + \text{KL}(B(1/2), B(1/2 + \epsilon)))$$
$$\leq T\left(2\frac{\psi^2}{\omega(1-\omega)} + \frac{\epsilon^2}{(\frac{1}{2}+\epsilon)(\frac{1}{2}-\epsilon)}\right)$$
$$\leq \frac{1}{4}$$

where the last step is obtained by setting $\psi = \frac{1}{4}\sqrt{\frac{\omega(1-\omega)}{T}}$ and $\epsilon = \frac{1}{6}\sqrt{\frac{1}{T}}$. Thus, for every $j \in \{2, 3, 4\}$:

$$\mathbb{P}_{\boldsymbol{\nu}_j}\left(\sum_{t=1}^T x_t(a_3) \geq T\frac{\epsilon}{\epsilon + \rho}\right) \geq \frac{3}{4} - \delta.$$

Which implies, for every $\delta \in (0, \frac{1}{2})$:

$$\mathbb{E}_{\boldsymbol{\nu}_j}\left[\sum_{t=1}^T x_t(a_3)\right] \geq \frac{1}{4}T\frac{\epsilon}{\epsilon + \rho}. \tag{13}$$

Now, we focus on instance $\boldsymbol{\nu}_2$ and $\boldsymbol{\nu}_3$: in the former the optimal strategy is to always pull $a_1$, while in the latter to always pull $a_2$. Hence, we can compute the expected regrets as:

$$\begin{cases} 2\mathbb{E}_{\boldsymbol{\nu}_2}[R_T] = \mathbb{E}_{\boldsymbol{\nu}_2}\left[\sum_{t=1}^T (\omega + \psi - \psi x_t(a_1) + (\Delta - \psi)x_t(a_3))\right] - T\omega, \\ 2\mathbb{E}_{\boldsymbol{\nu}_3}[R_T] = \mathbb{E}_{\boldsymbol{\nu}_3}\left[\sum_{t=1}^T (\omega + \psi - \psi x_t(a_2) + (\Delta - \psi)x_t(a_3))\right] - T\omega, \end{cases}$$

$$\begin{cases} 2\mathbb{E}_{\boldsymbol{\nu}_2}[R_T] = T\psi - \psi\mathbb{E}_{\boldsymbol{\nu}_2}\left[\sum_{t=1}^T (x_t(a_1) + x_t(a_3))\right] + \Delta\mathbb{E}_{\boldsymbol{\nu}_2}\left[\sum_{t=1}^T x_t(a_3)\right], \\ 2\mathbb{E}_{\boldsymbol{\nu}_3}[R_T] = T\psi - \psi\mathbb{E}_{\boldsymbol{\nu}_3}\left[\sum_{t=1}^T (x_t(a_2) + x_t(a_3))\right] + \Delta\mathbb{E}_{\boldsymbol{\nu}_3}\left[\sum_{t=1}^T x_t(a_3)\right]. \end{cases}$$

We now leverage Lemma A.1 from [Auer et al., 2002] and relate the expectations $\mathbb{E}_{\boldsymbol{\nu}_j}$, for $j \in \{2, 3\}$, with $\mathbb{E}_{\boldsymbol{\nu}_4}$:

$$\mathbb{E}_{\boldsymbol{\nu}_j}\left[\sum_{t=1}^T x_t(a_i)\right] \leq \mathbb{E}_{\boldsymbol{\nu}_4}\left[\sum_{t=1}^T x_t(a_i)\right] + T\sqrt{\frac{\ln 2}{2}\text{KL}_T(\boldsymbol{\nu}_4, \boldsymbol{\nu}_j)}, \tag{14}$$

for every $a_i$, where $\text{KL}_T$ indicates the KL-between the probability measures after $T$ rounds of history, and can be bounded as[8]:

$$KL_T(\boldsymbol{\nu}_4, \boldsymbol{\nu}_j) \leq T\frac{\psi^2}{\omega(1-\omega)}. \tag{15}$$

It follows:

$$\begin{cases} 2\mathbb{E}_{\boldsymbol{\nu}_2}[R_T] = T\psi - \psi\mathbb{E}_{\boldsymbol{\nu}_4}\left[\sum_{t=1}^T (x_t(a_1) + x_t(a_3))\right] - \sqrt{\frac{\ln 2}{2}\frac{\psi^4 T^3}{\omega(1-\omega)}} + \Delta\mathbb{E}_{\boldsymbol{\nu}_2}\left[\sum_{t=1}^T x_t(a_3)\right], \\ 2\mathbb{E}_{\boldsymbol{\nu}_3}[R_T] = T\psi - \psi\mathbb{E}_{\boldsymbol{\nu}_4}\left[\sum_{t=1}^T (x_t(a_2) + x_t(a_3))\right] - \sqrt{\frac{\ln 2}{2}\frac{\psi^4 T^3}{\omega(1-\omega)}} + \Delta\mathbb{E}_{\boldsymbol{\nu}_3}\left[\sum_{t=1}^T x_t(a_3)\right], \end{cases}$$

$$\begin{cases} 2\mathbb{E}_{\boldsymbol{\nu}_2}[R_T] = \psi\mathbb{E}_{\boldsymbol{\nu}_4}\left[\sum_{t=1}^T x_t(a_2)\right] - \sqrt{\frac{\ln 2}{2}\frac{\psi^4 T^3}{\omega(1-\omega)}} + \Delta\mathbb{E}_{\boldsymbol{\nu}_2}\left[\sum_{t=1}^T x_t(a_3)\right], \\ 2\mathbb{E}_{\boldsymbol{\nu}_3}[R_T] = \psi\mathbb{E}_{\boldsymbol{\nu}_4}\left[\sum_{t=1}^T x_t(a_1)\right] - \sqrt{\frac{\ln 2}{2}\frac{\psi^4 T^3}{\omega(1-\omega)}} + \Delta\mathbb{E}_{\boldsymbol{\nu}_3}\left[\sum_{t=1}^T x_t(a_3)\right], \end{cases}$$

---

[8]Note that KL divergence is invariant to scaling, thus we directly compare Bernoulli distribution instead of the distributions derived by dividing by two.

where the first step is a consequence of Equation (15), and the second step follows from the identity $x_t(a_1) + x_t(a_2) + x_t(a_3) = 1$, for every $t \in [T]$. We are now ready to lower bound the average expected regret between instances $\boldsymbol{\nu}_2$ and $\boldsymbol{\nu}_3$.

$$\mathbb{E}_{\boldsymbol{\nu}_2}[R_T] + \mathbb{E}_{\boldsymbol{\nu}_3}[R_T] \geq \psi \mathbb{E}_{\boldsymbol{\nu}_4}\left[\sum_{t=1}^{T}(x_t(a_1) + x_t(a_2))\right] - \sqrt{\frac{\ln 2}{2}\frac{\psi^4 T^3}{\omega(1-\omega)}} + \Delta\mathbb{E}_{\boldsymbol{\nu}_3}\left[\sum_{t=1}^{T}x_t(a_3)\right]$$

$$= \frac{T\psi}{2} + \left(\Delta - \frac{\psi}{2}\right)\mathbb{E}_{\boldsymbol{\nu}_4}\left[\sum_{t=1}^{T}x_t(a_3)\right] - \sqrt{\frac{\ln 2}{2}\frac{\psi^4 T^3}{\omega(1-\omega)}}$$

$$\geq \frac{T\psi}{2} + \frac{T\Delta}{8}\frac{\epsilon}{\epsilon + \rho} - \sqrt{\frac{\ln 2}{2}\frac{\psi^4 T^3}{\omega(1-\omega)}}$$

$$\geq \frac{1}{16}\sqrt{\omega(1-\omega)T} + \frac{1}{48\rho}\Delta\sqrt{T}$$

where the second inequality follows from Equation (13) and the fact that $\Delta \geq \psi$, the last inequality from the definition of $\psi$ and $\epsilon$, and the fact that $\rho \geq \epsilon$. Noting that $\max\{\mathbb{E}_{\boldsymbol{\nu}_2}[R_T], \mathbb{E}_{\boldsymbol{\nu}_3}[R_T]\} \geq (\mathbb{E}_{\boldsymbol{\nu}_2}[R_T] + \mathbb{E}_{\boldsymbol{\nu}_3}[R_T])/2$, we can conclude first step.

**Step 2** Note that $\frac{\Delta}{2} = \sqrt{\left(\frac{W_t + \Delta}{2} - \frac{W_t}{2}\right)^2}$, thus the (deterministic) term $\frac{\Delta}{2}\sqrt{T}$ is equivalent to the quadratic term describing the average squared distance between the optimal policy and the strictly safe strategy in instances $\boldsymbol{\nu}_2$ and $\boldsymbol{\nu}_3$.

Consider $T \geq \max\left\{2, (11 + \ln T)\left(\frac{8}{3}\right)^2\right\}$, and $\frac{1}{2} \geq \omega \geq \frac{1}{T}\max\left\{1, \left(\frac{11}{2} + \ln T\right)\right\}$. Note that the set of existence of $\omega$ is never empty due to the condition on $T$.

Consider modified versions of these instances with $\widetilde{\omega} = \frac{\omega}{2}$: we apply Bernstein Inequality to obtain that, with probability at least $1 - \delta'$:

$$\sum_{t=1}^{T}\boldsymbol{\ell}_t^\top \boldsymbol{x}^* = \sum_{t=1}^{T}\frac{W_t}{2}$$

$$\leq \frac{T\widetilde{\omega}}{2} + \frac{1}{2}\sqrt{2T\widetilde{\omega}(1-\widetilde{\omega})\ln\left(\frac{1}{\delta'}\right)} + \frac{1}{3}\ln\left(\frac{1}{\delta'}\right)$$

$$\leq \frac{T\omega}{4} + \frac{T\omega}{4} \leq \frac{T\omega}{2}, \tag{16}$$

by observing that $\left(\frac{8}{3}\right)^2\ln\left(\frac{1}{\delta'}\right) \leq T\omega$, which holds true for $\delta' = \frac{1}{228T}$, given the conditions on $T$ and $\omega$. Thus, instances $\boldsymbol{\nu}_2$ and $\boldsymbol{\nu}_3$ may generate, with high probability, loss sequences in which the optimal one is bounded by $\frac{T\omega}{2}$. We conclude the proof by showing that, among these loss sequences, at least one satisfies the previously derived lower bound. Without loss of generality, consider $\boldsymbol{\nu}_2$ to be the instance with the higher expected regret, we then apply the lower bound derived in Step 1 with $\widetilde{\omega}$ to get

$$\mathbb{E}_{\boldsymbol{\nu}_2}[R_T(\boldsymbol{\ell}_{1:T})] \geq \frac{9}{1024}\sqrt{T\omega} + \frac{1}{48\rho}\Delta\sqrt{T}, \tag{17}$$

where $\boldsymbol{\ell}_{1:T}$ is a fixed loss sequence generated $\boldsymbol{\nu}_2$, and using that $\omega \leq \frac{1}{2}$. Suppose by contradiction that, for every $\boldsymbol{\ell}_{1:T}$ s.t. Equation (16) holds, then

$$\mathbb{E}_{\boldsymbol{\nu}_2}[R_T(\boldsymbol{\ell}_{1:T})\mathbb{1}_{Eq.(16)}] < \frac{9}{2048}\sqrt{T\omega} + \frac{1}{48\rho}\Delta\sqrt{T},$$

Then, by setting $\delta' = \frac{1}{228T}$, we have:

$$\mathbb{E}_{\boldsymbol{\nu}_2}[R_T(\boldsymbol{\ell}_{1:T})] = \mathbb{E}_{\boldsymbol{\nu}_2}[R_T(\boldsymbol{\ell}_{1:T})\mathbb{1}_{Eq.(16)}] + \mathbb{E}_{\boldsymbol{\nu}_2}[R_T(\boldsymbol{\ell}_{1:T})(1 - \mathbb{1}_{Eq.(16)})]$$

$$< \frac{9}{2048}\sqrt{T\omega} + \frac{1}{48\rho}\Delta\sqrt{T} + \frac{1}{228}$$

$$< \frac{9}{1024}\sqrt{T\omega} + \frac{1}{48\rho}\Delta\sqrt{T},$$

where the first inequality derives from bounding the regret with $T$ and the definition of $\delta'$, and the second inequality derives from observing that $T\omega > \left(\frac{2048}{9\cdot228}\right)^2$, given the conditions on $T$ and $\omega$. This represents a contradiction with the lower bound derived in the first step, and thus there must exist a loss sequence such that Equation (16) and Equation (17) hold simultaneously. □

