# OpenReview forum: "Data-Dependent Regret Bounds for Constrained MABs"
_NeurIPS.cc/2025/Conference — NeurIPS 2025 poster_

### Official Review · Reviewer_eMxG · 2025-06-29

**Clarity:** 3
**Significance:** 3
**Originality:** 3
**Rating:** 5
**Confidence:** 3

**Summary:**

The paper studies the multi-armed-bandit (MAB) problem where the learner chooses an arm among K fixed choices at each round t from 1 to T. At each t the learner receives a reward associated to the chosen arm and strives to maximise the cumulated sum of rewards. The paper considers adversarial rewards, and so good strategies are randomized and actually need to choose a vector of probabilities for each arm, at each round.

While this problem is very classical, the specificity of the paper is consider constraints. At each round there are realizations (iid over rounds) of constraints, and the probabilities chosen by the learner must satisfy the constraints, either by having a sublinear cumulated sum of violations, or by having no violation with high probability.

The paper studies the regret, defined as the difference between an optimal strategy in hindsight that satisfies the constraints, and the cumulated sum of rewards obtained by the learner. The goal is to provide bounds that are called "data-dependent", that is that depend on the actual adversarial sequence of rewards. These bounds are more informative than worst-case bounds.

The paper provides an algorithm called SOLB (safe OMD with log barrier) and prove an upper bound for its regret that is data-dependent and is interpreted as the sum of a safety-complexity term and a bandit-complexity term. The paper provides a lower bound as a counterpart, that also has these two terms.

**Questions:**

Remark 2 uses a lower bound from 2016 for unconstrained MABs. I was surprised that the remark writes that, thus, an upper bound for constrained MABs is tight. Would there be a restricted class of constrained MABs that are "far from unconstrained ones", and for which it would be an open question to provide a lower bound?

Could a discussion of concrete motivations for studying constrained MABs be added to the paper. Many readers can have practical motivations in mind for unconstrained MABs (say, for online advertisment) but I think many readers would benefit from examples of concrete constraints.


Line 332, is it the fifth term rather than the fourth?

**Ethical Concerns:**

["NO or VERY MINOR ethics concerns only"]

**Final Justification:**

Given the author's response to me, and to the other reviews, I am quite confident in my positive evaluation of the paper, in the end.

**Limitations:**

YES

**Paper Formatting Concerns:**

No concerns

**Quality:**

3

**Strengths And Weaknesses:**

STRENGTHS The paper is well-written. Although the content is technical, the authors have made an effort to provide intuition, along the exposition. The topic is important for machine learning theory. The contribution is strong, with a main informative data-dependent regret bound, that is globally matched by a lower bound. The upper bound required to design a new algorithm. The proofs appear to have required a lot of work and appear to be theoretically interesting.


WEAKNESSES I would prefer not to point out a specific weakness, as I do not have notable weaknesses in mind for the paper.

---

> ### Author Rebuttal · Authors · 2025-07-30
>
> We thank the Reviewer for the positive evaluation of our work.
>
> > Remark 2 uses a lower bound from 2016 for unconstrained MABs. I was surprised that the remark writes that, thus, an upper bound for constrained MABs is tight. Would there be a restricted class of constrained MABs that are "far from unconstrained ones", and for which it would be an open question to provide a lower bound?
>
> Our algorithm, COLB, matches the existing regret lower bound for the unconstrained settings. However, we remark that our algorithm also exhibits an upper bound on the cumulative violations. This ''double'' guarantee differentiates the performance analysis of MAB algorithms in the soft constraints setting from the ones in the unconstrained setting. It doesn't matter how much constrained the instance is; the worst-case upper bound on the regret is still in that order of magnitude, as well as the bound on the cumulative violations. This is due to the worst-case nature of the result.
>
> > Could a discussion of concrete motivations for studying constrained MABs be added to the paper. Many readers can have practical motivations in mind for unconstrained MABs (say, for online advertisment) but I think many readers would benefit from examples of concrete constraints.
>
> In the final version of the paper, we will add the following discussion about applications to online advertising. In an online advertisement scenario, from the perspective of a publisher, multiple ads (say, $K$ distinct ads) are available to be displayed to incoming users. Every ad is associated with an expected revenue, which may depend on the type of ad or its position. In an unconstrained scenario, the problem would just be to find the most profitable ad strategy. However, displaying an ad is associated with an economic cost (the spend-per-impression), and a marketing campaign has limited funding. In this sense, a publisher trying to maximize its profits in an online manner resorts to the constrained MAB model, with either soft or hard constraints, depending on its own spending policy.
>
> > Line 332, is it the fifth term rather than the fourth?
>
> We thank the Reviewer for spotting the typo. We will surely modify it in the final version of the paper.

---

> > ### Comment · Reviewer_eMxG · 2025-08-01
> > **Response acknowledgement**
> >
> > I thank the reviewers for their response and clarifications.
> > I am happy to maintain my positive evaluation of the paper.

---

### Official Review · Reviewer_5KbZ · 2025-07-01

**Clarity:** 3
**Significance:** 2
**Originality:** 2
**Rating:** 4
**Confidence:** 4

**Summary:**

Constrained bandits represent a compelling and practically relevant problem, motivated by numerous real-world applications. While the existing rich literature primarily focuses on horizon-dependent guarantees, this paper initiates the study of **data-dependent guarantees** in the setting of **adversarial losses** and **stochastic constraints**. The authors make the following key contributions:
1) **COLB** : An algorithm designed for the soft constraint setting, where the learner is allowed to violate constraints. The authors provide a classical upper bound on constraint violations and a data-dependent upper bound on the regret.
2) **SOLB** : An algorithm tailored for the hard constraint setting, where the learner must satisfy the constraints at each round with high probability. The authors prove that the algorithm does not violate the constraints and provide a data-dependent upper bound on the regret.
3) **Lower Bound**: The paper also presents a lower bound that supports the tightness of the proposed upper bounds, reinforcing the theoretical soundness of the results.

**Questions:**

**Questions and remarks:**
1) Please address the weaknesses mentioned above.
2) In the proof of Lemma 2, particularly the first inequality following line 546, it is unclear how the bound holds uniformly for all $t$ without any additional assumption on the norm of $\hat{g}_{t,i} - \beta_t$. Could you please elaborate on this step?
3) In the proof of Lemma 5, you use the inequality $\kappa^{n_a} \leq 5$. Could you please explain how this bound is derived?
4) In the proof of Theorem 2, could you please clarify the inequality that appears after Line 586? It seems to me that $\hat{l}_t$ is sparser than $l_t - \hat{l}_t$.
5) I suggest modifying the definition of $V_T$ by placing the '$\max$' operator inside the summation. This adjustment would ensure that there is no compensation between rounds across different arms. Moreover, the theoretical proofs provided in the paper would remain exactly the same under this modification.
6) Can you please share your opinion on the case where the losses (or rewards) are stochastic while the constraints are adversarial?

**Minor typos:**
1) In Line 13 of the SOLB algorithm, should it be $\tilde{x_{t+1}}$ instead of $x_{t+1}$?
2) In the proof of Lemma 2, the second equality following line 548 appears to reference the set $\tilde{S}$. However, I believe it should be $S^{\circ}$ instead, based on the context and the structure of the argument.


I am fully open to revising my evaluation and raising my score if the issues I raised are adequately addressed.

**Ethical Concerns:**

["NO or VERY MINOR ethics concerns only"]

**Final Justification:**

I have read the authors’ rebuttal as well as the discussion from other reviewers. I find the paper to be well written and easy to follow, with a high level of mathematical rigor. The work addresses a novel and previously unexplored question, although it does so by adapting well-established techniques from the literature (such as Online Mirror Descent and algorithmic frameworks for bandits with constraints).

Overall, I am positive about the paper and believe it makes a contribution to the field.

**Limitations:**

yes

**Quality:**

3

**Strengths And Weaknesses:**

**Strengths:**
1) The paper is well written and easy to follow.
2) The problem addressed is interesting and opens a promising direction for research.
3) The mathematical analysis appears to be sound.

**Weaknesses:**
1) While the question of data-dependent guarantees for constrained MABs is novel, both data-dependent regret bounds for unconstrained MABs and algorithmic design for constrained MABs have been extensively studied. This paper appears to combine several previously proposed techniques and results. In particular, the OMD with log-barrier algorithm introduced by [lee et al. 2020] is heavily utilized in the proposed algorithm, especially in the analysis techniques, which are central to the proofs of the main theorems. Additionally, the structure of the feasible sets and the combination factor idea bear strong resemblance to those presented in [Pacchiano et al. 2021] and [Amani et al. 2019] .
2) I have an issue with the paragraph titled **Safe Decision Space** (Lines 152--160). I am unable to see how, by invoking Lemma 1, one can conclude that selecting a strategy from $S_t$ ensures satisfaction of the constraints with high probability. Specifically, the definition of $S_t$ involves a subtraction, and I believe the claim would only hold if the definition used a '+' instead of a '-' sign, or alternatively, if the constraints were of the form $g_i^\top x \geq \alpha_i$. As it stands, the justification for the safety guarantee is unclear.
3) The quantity $D_{\psi_{t}}$ is used extensively throughout the algorithms. However, unless I missed it, it does not appear to be explicitly defined in the main text.
4) It would strengthen the paper to include numerical simulations that empirically validate the theoretical claims. Currently, the paper lacks such experiments, which are essential for demonstrating the practical relevance and performance of the proposed algorithms.

---

> ### Author Rebuttal · Authors · 2025-07-30
>
> We thank the Reviewer for the positive evaluation of our work.
>
> > While the question of data-dependent guarantees for constrained MABs is novel, both data-dependent regret bounds for unconstrained MABs and algorithmic design for constrained MABs have been extensively studied. This paper appears to combine several previously proposed techniques and results. In particular, the OMD with log-barrier algorithm introduced by [lee et al. 2020] is heavily utilized in the proposed algorithm, especially in the analysis techniques, which are central to the proofs of the main theorems. Additionally, the structure of the feasible sets and the combination factor idea bear strong resemblance to those presented in [Pacchiano et al. 2021] and [Amani et al. 2019] .
>
> We certainly agree with the Reviewer on the fact that our algorithm for the soft constraints setting relies on existing algorithms that attain data-dependent bounds in unconstrained settings.
>
> Nonetheless, we believe that our results contain technical novelty and may be of independent interest. Both the regret upper bound and the regret lower bound require a novel (and challenging) analysis w.r.t. state-of-the-art unconstrained settings. This is a consequence of the fact that the (A) term and the (B) term must be treated separately (in both upper and lower regret bounds), preventing the use of existing techniques.
>
> Moreover, while our techniques to handle the constraints satisfaction may share similarities with [Pacchiano et al. 2021] and [Amani et al. 2019], we underline that their techniques cannot be generalized to our setting, due to the adversariality of the losses. Indeed, their approach is summarized as follows. They play the strictly safe strategy for a certain number of rounds, then they play on a pessimistic safe set. We cannot adopt their approach, since adversarial no-regret algorithms do not work on enlarging decision spaces, such as pessimistic ones. Instead, we optimize on an optimistic decision space (that is, a decreasing one) and then perform a convex combination on the strictly safe strategy.
>
> > I have an issue with the paragraph titled Safe Decision Space (Lines 152--160).
>
> We thank the Reviewer for the opportunity to clarify this aspect. We believe that there is a possible misunderstanding. Playing in $S_t$ at each round allows us to obtain **sublinear cumulative violation**. The safety property is attained by an additional convex combination with the safe known strategy.
>
> From a technical perspective, lower confidence bounds are employed since the safe optimum must be included in $S_t$ at each $t$; thus, it is necessary to be optimistic on the constraints satisfaction, which results in employing the lower confidence bound. Then, it is easy to show that $S_t$ concentrates on the true safe set at a rate of $1/\sqrt{T}$, resulting in a $\widetilde{\mathcal{O}}(\sqrt{T})$ violation bound.
>
> We will better specify it in the final version of the paper.
>
> Since this is a crucial aspect, please let us know if further discussion is necessary.
>
> > The quantity $D_{\psi_t}$ is used extensively throughout the algorithms. However, unless I missed it, it does not appear to be explicitly defined in the main text.
>
> We thank the Reviewer for the comment. We forgot to explicitly state the Bregman divergence definition, since this directly follows from the regularizer $\psi_t$ definition. Nonetheless, we will explicitly specify it in the final version of the paper.
>
> > In the proof of Lemma 2, particularly the first inequality following line 546, it is unclear how the bound holds uniformly for all  $t$ without any additional assumption on the norm of $\hat g_t-\beta_t$. Could you please elaborate on this step?
>
> We thank the Reviewer for the comment. The Reviewer is correct. In general, in UCB approaches, when the rewards are bounded in $[0,1]$, the quantity (empirical mean of the) rewards plus concentration bound is capped to $1$. Similarly, in our case, we do the same for the constraints, taking $0$, when $\hat g_t -\beta_t$ is negative.
>
> We will better clarify it in the final version of the paper.
>
> > In the proof of Lemma 5, you use the inequality $k^{n_a}\leq 5$. Could you please explain how this bound is derived?
>
> $\kappa$ is defined as $e^{1/\ln T}$. We can upper bound $n_a$ with $\ln 5 \ln T$, noticing that $x_t(a)$ is lower bounded by $1/T$ and initialized to $1/K$ and $h_a$ is initialized as $2K$ and updated as $2/x_t(a)$ every time $1/x_t(a)>h_a$.
>
> > In the proof of Theorem 2, could you please clarify the inequality that appears after Line 586?
>
> The Reviewer is correct in stating that $\hat{\boldsymbol{\ell}}\_t$ is sparser than $\boldsymbol{\ell}$, even if $\hat{\boldsymbol{\ell}}\_t$ is larger than $\boldsymbol{\ell}$, in terms of $||\cdot\||_{\infty}$ norm. Nonetheless, the bound (at the end of the inequalities) trivially holds by bounding the difference with $\mathbb{E}_t[(\boldsymbol{\ell}_t^\top \boldsymbol{x}_t)^2]$. Thus, in the proof, we focus on the term $\mathbb{E}_t[(\widehat{\boldsymbol{\ell}}_t^\top \boldsymbol{x}_t)^2]$, which needs more explanation. In the final version of the paper, we will better specify this step. We thank the Reviewer for spotting this aspect.
>
>
>
> > I suggest modifying the definition of $V_T$ by placing the $\max$ operator inside the summation. This adjustment would ensure that there is no compensation between rounds across different arms. Moreover, the theoretical proofs provided in the paper would remain exactly the same under this modification.
>
> We thank the Reviewer for the comment. We employed our definition of $V_T$ since it is standard in the literature. Nonetheless, we do not understand how placing the $\max_{i\in[m]}$ operator inside the summation would prevent compensation between arms. Indeed, the compensation between rounds is prevented by the $[\cdot]^+$ operator, while the definition of violation is with respect to the strategy, and not with respect to a specific action.
>
> > Can you please share your opinion on the case where the losses (or rewards) are stochastic while the constraints are adversarial?
>
> The impossibility result on learning with adversarial constraints carries on when the losses are stochastic, too. Technically, it still holds for deterministic fixed losses. We refer to [1] for additional details.
>
> [1] Online learning with sample path constraints. Shie Mannor, John N. Tsitsiklis, and Jia Yuan Yu (2009)
>
> > In Line 13 of the SOLB algorithm
>
> We thank the Reviewer for pointing out the typo, we will correct it in the final version of the paper.
>
> > In the proof of Lemma 2, the second equality following line 548 appears to reference the set $\tilde S$. However, I believe it should be $S^\circ$ instead, based on the context and the structure of the argument.
>
> We thank again the Reviewer for pointing out the typo.  We will correct the typo in the final version of the paper.

---

> > ### Comment · Reviewer_5KbZ · 2025-08-02
> >
> > I thank the reviewer for their response. I would like to reiterate that numerical experiments are valuable, as they demonstrate the practical relevance and performance of the proposed algorithms. I strongly encourage the authors to include such experiments in the final version of the paper. I maintain my positive score.

---

### Official Review · Reviewer_UfDE · 2025-07-02

**Clarity:** 2
**Significance:** 3
**Originality:** 2
**Rating:** 4
**Confidence:** 3

**Summary:**

This paper studies data-dependent regret bounds in constrained Multi-Armed Bandit (MAB) seetings. The authors focus on the scenario of adversarial losses and stochastic hard constraints (due to the impossibility result for fully adversarial scenarios). They propose two algorithms, which are Constrained OMD with Log-Barrier (COLB) for soft constraints and the main algorithm Safe OMD with Log-Barrier (SOLB) for hard constraints. SOLB satisfies the constraints with high probability by dynamically combining and OMD-suggested strategy with a strictly feasible one. They show that the high probability regret bound has two data-dependent terms: a safety complexity term that quantifies the difficulty introduced by satisfying the constraints; and a bandit complexity term that represents the learning complexity independent of the constraints. The authors also provide a lower bound to demonstrate that these two terms are in fact fundamental and inherent characteristics of the problem and the proposed algorithm SOLB is optimal up to logarithmic terms.

**Questions:**

1) Can you possibly relax (or maybe adaptively learn) the strictly feasible strategy and its associated costs? It would be nice to discuss this even as a future research direction.

2) Can you give a more in-depth explanation for your conjecture that the $1/\rho$ dependence should affect only the Safety Complexity term (A), and not the Bandit Complexity term (B)? What specific challenges exist in your derivations in achieving a tighter bound?

3) The learning rate for COLB and SOLB requires knowledge of the optimal cumulative loss. While "doubling trick" is mentioned, how would this trick specifically be applied?

4) What are the hypothetical practical applications where your algorithm and results would be most critical and beneficial?

**Ethical Concerns:**

["NO or VERY MINOR ethics concerns only"]

**Final Justification:**

I am keeping my score, which remains positive.

The algorithms (COLB, SOLB) are novel, addressing a previously overlooked area and providing an affirmative answer to the research question about data-dependent regret bounds in constrained MABs. The decomposition of regret into "Safety Complexity" (difficulty of satisfying constraints) and "Bandit Complexity" (complexity of learning independently of constraints) offers valuable insights. A lower bound shows these terms are fundamental, and SOLB's bound is optimal up to logarithmic factors. The authors confirmed that a standard "doubling trick" can handle the requirement of knowing the optimal cumulative loss, incurring only negligible logarithmic factors while maintaining safety guarantees. A hypothetical example in online advertisement was provided, offering context for the algorithms.

SOLB requires explicit knowledge of a strictly feasible strategy and its associated costs as input. While theoretically justified to prevent impossibility results and ensure exploration, this assumption limits broader applicability where such a strategy might not be readily available. The paper is purely theoretical, which is acceptable but means practical insights lack empirical support.

The strong theoretical contributions and proofs (novelty, decomposition, tightness) heavily outweigh the identified practical limitations and remaining subtle theoretical questions.

**Limitations:**

Yes

**Paper Formatting Concerns:**

No major issues.

**Quality:**

2

**Strengths And Weaknesses:**

## Strengths

1) The algorithms seem novel. COLB for soft constraints serves as a warm-up by achieving small-loss regret bounds and the primary contribution SOLB provides optimal regret for hard constraints.

2) The theoretical insights on the data-dependent regret bound, especially the natural interpretation of the two independent terms safety and bandit complexity is very nice.

3) The regret bound is optimal (up to logarithmic factors).

## Weaknesses

1) SOLB requires Slater's condition and explicit knowledge of a strictly feasible strategy and its associated costs as input, which seems limiting, especially in practical applications.

2) The initial setting of the learning rate requires knowledge of the optimal cumulative loss.

3) While not a major weakness, there are no empirical validation.

---

> ### Author Rebuttal · Authors · 2025-07-30
>
> We thank the Reviewer for the positive evaluation of our work.
>
> > SOLB requires Slater's condition and explicit knowledge of a strictly feasible strategy and its associated costs as input, which seems limiting, especially in practical applications.
>
> We thank the Reviewer for the opportunity to further elaborate on the assumptions. **Assumption 1.** Assumption 1 (namely $\rho>0$) is provably necessary in the hard constraints setting, as highlighted by Theorem 5. To see this, notice that, when Assumption 1 does not hold, a strategy that is arbitrarily close to the safe one (that is, the strategy with $\rho=0$) could violate the constraints, thus preventing any form of exploration.
> We underline that Slater's condition is commonly accepted in the constrained online learning literature.
> **Assumption 2.** Assumption 2 states that the algorithm is given a strictly feasible solution (and its costs).
> From a theoretical perspective, this assumption allows the algorithm to simultaneously explore and satisfy the constraints when no information on the environment is available, that is, in the very first rounds. For instance, in the first round, the algorithm has to play randomly; Assumption 2 allows to be safe in the first round while possibly ``exploring" of a $\rho$ factor.  **From a practical perspective**, we believe that this requirement is met in many real-world interesting scenarios, where a void action, that is, an action with zero constraint violation and zero reward, is available to the learner (e.g., online auctions, where the bidder may bid $0$ whenever its budget is depleted).
>
> > The initial setting of the learning rate requires knowledge of the optimal cumulative loss. The learning rate for COLB and SOLB requires knowledge of the optimal cumulative loss. While "doubling trick" is mentioned, how would this trick specifically be applied?
>
> The doubling trick we mention is the one developed in [1] (Appendix C.3) and [2] (Remark 1). In [1], the authors introduce such a doubling trick for adapting to the very same quantity. Their setting involves graph-feedback, but one can plug an unconnected graph and resort to a standard bandit feedback, recovering the exact algorithmic steps.
>
> [1] Chung-Wei Lee, Haipeng Luo, and Mengxiao Zhang. A closer look at small-loss bounds for bandits with graph feedback. In Conference on Learning Theory, 2020.
>
> [2] Lee, C. W., Luo, H., Wei, C. Y., Zhang, M. (2020). Bias no more: high-probability data-dependent regret bounds for adversarial bandits and mdps. Advances in neural information processing systems, 33, 15522-15533.
>
> > Can you possibly relax (or maybe adaptively learn) the strictly feasible strategy and its associated costs? It would be nice to discuss this even as a future research direction.
>
> We believe that it is possible to dynamically learn the Slater's parameter and the strictly feasible strategy on the fly, paying an additional constant term (independent of $T$) in both regret and violation, that is, **the safety property is not attainable** but constant violation is. This probably can be done employing, for a constant number of rounds, a primal-dual Lagrangian approach on the constraints only, so that the optimal solution becomes the safest one corresponding to Slater's parameter. Then, it is sufficient to run our algorithm for the hard constraints with the estimates attained in the previous phase.
>
> We will include this discussion in the final version of the paper.
>
> > Can you give a more in-depth explanation for your conjecture that the  $1/\rho$ dependence should affect only the Safety Complexity term (A), and not the Bandit Complexity term (B)? What specific challenges exist in your derivations in achieving a tighter bound?
>
> We thank the Reviewer for the interesting question. From a high-level perspective, the (B) term arises from the intrinsic hardness of attaining small loss regret bounds. Differently, the (A) term arises from the safety property, which forces any algorithm to play the strictly feasible solution a number of times that depends on $\rho$. Thus, if we consider unconstrained MABs, which are the trivial case of constrained MABs, the (A) term disappears, and the regret bound consists of the second term only. Thus, since the $\rho$ terms do not appear in unconstrained cases, it is reasonable that it is linked to the (B) term only.
>
> As concerns the technical challenges to achieve a tighter bound, they are summarized in the following. The main idea we had to prove the tight bound was to split the analysis of Lemma 4 in two phases: (i) when the convex combination parameter $\gamma_t$ is large and (ii) when the convex combination parameter is small enough---notice that a similar approach is employed in the current analysis of Lemma 3, to avoid a $1/\rho^2$ dependence---, and proving that $\widetilde R_T$ in the first phase ($\gamma_t$ large) is constant while $\widetilde R_T$ in the second phase is independent on $\rho$. There are two key challenges in doing that. First, it is necessary to show that after a constant number of rounds, $\tilde{\boldsymbol{x}}_t$ is not to far from the uniform strategy (that is, the initialization). This can be done only assuming that $T$ is very large. Second, it is not true that there exists a round $\bar t$ which splits exactly the two phases. Thus, the analysis may result inconsistent in adversarial settings, since the rounds must be consecutive to properly bound the "regret", or, in this case, $\widetilde R_T$.
>
> Please let us know if further discussion is necessary.
>
> > What are the hypothetical practical applications where your algorithm and results would be most critical and beneficial?
>
> In the following, we provide a practical application in the context of online advertisement. In an online advertisement scenario, from the perspective of a publisher, multiple ads (say, $K$ distinct ads) are available to be displayed to incoming users. Every ad is associated with an expected revenue, which may depend on the type of ad or its position. In an unconstrained scenario, the problem would just be to find the most profitable ad strategy. However, displaying an ad is associated with an economic cost (the spend-per-impression), and a marketing campaign has limited funding. In this sense, a publisher trying to maximize its profits in an online manner resorts to the constrained MAB model, with either soft or hard constraints, depending on its own spending policy.
>
> We will surely include this discussion in the final version of the paper.

---

> > ### Comment · Reviewer_UfDE · 2025-08-05
> >
> > I appreciate the authors' rebuttal and have some follow up questions.
> >
> > - You point out that the initial learning rate's dependence on the optimal cumulative loss can be relaxed using a "doubling trick". While this is a standard technique, could you discuss any specific implications or trade-offs that arise when applying the doubling trick in the context of hard constraints? For instance, does it introduce any subtle challenges in maintaining the high-probability constraint satisfaction guarantee, or does it primarily affect constant factors in the regret bound?
> >
> > - In the online advertisement example, you explain how the problem aligns with the constrained MAB framework. Could you elaborate on how the data-dependent nature of SOLB's regret bounds provides a tangible benefit in this specific advertising scenario compared to a standard $\tilde{O}(\sqrt{T})$ bound? For instance, when would the "small-loss" characteristic of your bound lead to a significantly better outcome for the publisher, and what problem instance characteristics would make this difference most pronounced?

---

> > > ### Author Response · Authors · 2025-08-06
> > >
> > > We thank the Reviewer for having appreciated our rebuttal.
> > >
> > > > You point out that the initial learning rate's dependence on the optimal cumulative loss can be relaxed using a "doubling trick". While this is a standard technique, could you discuss any specific implications or trade-offs that arise when applying the doubling trick in the context of hard constraints? For instance, does it introduce any subtle challenges in maintaining the high-probability constraint satisfaction guarantee, or does it primarily affect constant factors in the regret bound?
> > >
> > > We thank the Reviewer for the interesting question. The doubling trick does not affect the violation---similarly, it does not affect the high probability safety guarantee---while it simply adds negligible factors (up to logarithmic ones) to the regret bound. Intuitively, this happens since the constraints estimation does not require any doubling trick to be applied; thus, the constraints estimation is equivalent to the standard case in which the optimal cumulative loss is known. To conclude, the main trade-off of this adaptation strategy concerns an additional logarithmic term in the regret bound only.
> > >
> > > > In the online advertisement example, you explain how the problem aligns with the constrained MAB framework. Could you elaborate on how the data-dependent nature of SOLB's regret bounds provides a tangible benefit in this specific advertising scenario compared to a standard $\tilde{\mathcal{O}}(\sqrt T)$ bound? For instance, when would the "small-loss" characteristic of your bound lead to a significantly better outcome for the publisher, and what problem instance characteristics would make this difference most pronounced?
> > >
> > > Our data-dependent guarantees allow the regret to scale with the quantity $\sqrt{\sum_{t=1}^T \boldsymbol{\ell}_t^\top \boldsymbol{x}^*}$ instead of $\sqrt{T}$. The former quantity is always smaller than or equal to the latter, allowing for tighter regret bounds.
> > >
> > > In practice, this allows both COLB and SOLB to exploit ''easy'' instances, i.e., instances in which the optimal policy suffers a small cumulative loss. In the advertisement example, an instance in which our algorithms significantly outperform the existing ones is one in which at least one of the available ads performs really well, and it is thus possible to achieve a high total revenue over the whole campaign. In a business domain, this is important because our algorithms perform well when the stake is high, exploiting ''rich'' advertising campaigns. When there isn't much to gain, and the campaign leads to a small profit even with the best possible policy, our algorithms still perform better than or equal to the existing ones.
> > >
> > > This result departs from the standard worst-case perspective of the problem, and in fact, real-world instances are usually better than the theoretical worst-case ones. In the worst-case instances, however, our guarantees resort to the known ones that scale with $\sqrt{T}$ without deteriorating them.

---

### Official Review · Reviewer_JrAR · 2025-07-21

**Clarity:** 3
**Significance:** 3
**Originality:** 3
**Rating:** 4
**Confidence:** 3

**Summary:**

In this paper, the authors consider a constrained MAB formulation, where the goal is to provide data-dependent regret bounds. Here, losses are adversarial, but constraints are modelled as being stochastic. The authors propose an algorithm that adapts OMD in conjunction with 'safe' decision spaces.

**Questions:**

1. Can the authors provide an intuitive explanation of why lower confidence bounds (LCBs), rather an UCBs, of the arm constraints are used to define safe decision spaces? I am referring to the display equation after Line 155 on Page 4. Note that with high probability, the actual $g_i$ are higher than their LCB. So the following sentence that with high probability, the set $\mathcal{S}_t$ contains actions that satisfy the constraints also feels suspicious.

2. Line 11 of Algorithm 1: The function $D_{\psi_t}(x,x_t)$ is not defined anywhere as far as I can tell. The meaning of this does not follow from the definition of $\psi_t(x)$ that follows. A similar comment applies to Algorithm 2.

3. Note that regret calculations here include expectations with respect to algorithm randomisation. However, it feels clumsy to provide a \emph{high-probability} bound on the \emph{average} regret. Now, I understand the "high-probability" bit is due to the stochastic constraint modelling, and the "average" bit is over the algorithm randomisation. A brief commentary on this would be useful, IMO.

**Ethical Concerns:**

["NO or VERY MINOR ethics concerns only"]

**Limitations:**

Yes

**Quality:**

3

**Strengths And Weaknesses:**

One natural concern is whether it is reasonable to assume adversarial losses in conjunction with stochastic constraints. While the authors do point to an impossibility result to justify the assumption, it is not clear to me whether the proposed setting captures any application reasonably.

Moreover, the algorithms are assumed to have bounds on the loss of optimal action in hindsight, plus a strictly feasible strategy (Assumption 2). Again, it is perhaps true that the problem is intractable in the absence of such knowledge, but I wonder is what that  really means is that the problem is ill-posed.

Aside from the above reservations on the problem formulation itself, the paper seems to make useful algorithmic contributions. The decomposition of the regret into safety complexity and bandit complexity is also quite interesting.

---

> ### Author Rebuttal · Authors · 2025-07-30
>
> We thank the Reviewer for the positive evaluation of our work.
>
> > One natural concern is whether it is reasonable to assume adversarial losses in conjunction with stochastic constraints. While the authors do point to an impossibility result to justify the assumption, it is not clear to me whether the proposed setting captures any application reasonably.
>
> As the Reviewer noticed, the main motivation of our setting with adversarial losses and stochastic constraints is theoretical, as it is provably not possible to attain sublinear regret and violation when the constraints are adversarial as well. Nonetheless, we think that there are several applications where the loss (reward) function exhibits some sort of non-stationarity, and the constraints are instead sampled from fixed distributions. Indeed, losses (rewards) may be determined from events occurring in the external environment and possibly influenced by other learning agents (being thus possibly non-stationary), while in many applications, constraints encode some inherently stationary requirements, e.g., some resource consumption not affected by the external environment.
>
> > Moreover, the algorithms are assumed to have bounds on the loss of optimal action in hindsight, plus a strictly feasible strategy (Assumption 2). Again, it is perhaps true that the problem is intractable in the absence of such knowledge, but I wonder if what that really means is that the problem is ill-posed.
>
> We thank the Reviewer for the interesting question. For the bound on the loss of the optimal action, we underline that a standard doubling trick approach can easily remove this assumption, that is, the quantity can be estimated on the fly, paying an additional logarithmic factor in the regret [1,2]. As concerns Assumption 2, this is met in every application where there exists a void action that attains $0$ violation, such as online auctions, where the bidder may decide to bid the $0$ value, not incurring any cost. If knowledge on the strictly feasible strategy is not available, the problem becomes ''unlearnable'' according to the standard bandit definition, i.e., the worst-case regret suffered is linear.
>
> [1] Chung-Wei Lee, Haipeng Luo, and Mengxiao Zhang. A closer look at small-loss bounds for bandits with graph feedback. In Conference on Learning Theory, 2020.
>
> [2] Lee, C. W., Luo, H., Wei, C. Y., Zhang, M. (2020). Bias no more: high-probability data-dependent regret bounds for adversarial bandits and mdps. Advances in neural information processing systems, 33, 15522-15533.
>
> > Can the authors provide an intuitive explanation of why lower confidence bounds (LCBs), rather an UCBs, of the arm constraints are used to define safe decision spaces? I am referring to the display equation after Line 155 on Page 4. Note that with high probability, the actual $g_i$ are higher than their LCB. So the following sentence, that with high probability, the set $S_t$ contains actions that satisfy the constraints also feels suspicious.
>
> We thank the Reviewer for the comment. We believe that there is a possible misunderstanding. Playing in $S_t$ at each round allows us to obtain a sublinear cumulative violation. The safety property is attained by an additional convex combination with the safe known strategy.
>
> From a technical perspective, LCBs are employed since the safe optimum must be included in $S_t$ at each $t$; thus, it is necessary to be optimistic on the constraints satisfaction, which results in employing the lower confidence bound. Then, it is easy to show that $S_t$ concentrates on the true safe set at a rate of $1/\sqrt{T}$, resulting in a $\widetilde{\mathcal{O}}(\sqrt{T})$ violation bound.
>
> > Line 11 of Algorithm 1
>
> We thank the Reviewer for pointing out this aspect. We forgot to explicitly state the Bregman divergence definition, since this directly follows from the regularizer $\psi_t$ definition. Nonetheless, we will explicitly specify it in the final version of the paper.
>
> > Note that regret calculations here include expectations with respect to algorithm randomisation. However, it feels clumsy to provide a \emph{high-probability} bound on the \emph{average} regret. Now, I understand the "high-probability" bit is due to the stochastic constraint modelling, and the "average" bit is over the algorithm randomisation. A brief commentary on this would be useful, IMO.
>
> We thank the Reviewer for the interesting comment.  Our regret definition employs strategy mixtures in place of the actual actions played by the algorithm and compares them to the optimal mixed strategy. This is done to be consistent with the constrained online learning literature, where the optimum is not pure, but it is a distribution over the actions. We remark that we do not take the expectation over the algorithm's randomization in the regret definition: indeed, we use OMD with log-barrier to deal with the loss estimation without taking the expectation on the algorithm's randomization. Thus, the ''high probability'' also encompasses the algorithm's randomization.
>
> We will surely include this comment in the final version of the paper. Please let us know if further discussion is necessary.

---

> > ### Comment · Reviewer_JrAR · 2025-08-07
> > **Thank you**
> >
> > Thank you for the clarifications on my questions.
> > My (already somewhat positive) rating on this paper remains unchanged. With the authors the very best of luck.

---

### Decision · Program_Chairs · 2025-09-17

**Decision:**

Accept (poster)

**Comment:**

This paper considers the problem of deriving data-dependent small-loss type bounds for multi-armed bandits with constraints.  The authors give two algorithms for soft and hard constraints, COLB (soft) and SOLB (hard), and prove data-dependent regret bounds for them that decompose into “safety” and “bandit” complexities; the authors also give matching lower bounds which suggest that the regret bounds are (qualitatively) optimal. Reviewers were generally positive on the paper, and felt that it gives a relatively complete answer to a question that has not been explicitly explored in the existing literature.